# FACT modulates the conformations of histone H2A and H2B N-terminal tails within nucleosomes

Yasuo Tsunaka [1], Hideaki Ohtomo[1] & Yoshifumi Nishimura [1,2 ✉]

Gene expression is regulated by the modification and accessibility of histone tails within nucleosomes. The histone chaperone FACT (facilitate chromatin transcription), comprising SPT16 and SSRP1, interacts with nucleosomes through partial replacement of DNA with the phosphorylated acidic intrinsically disordered (pAID) segment of SPT16; pAID induces an accessible conformation of the proximal histone H3 N-terminal tail (N-tail) in the unwrapped nucleosome with FACT. Here, we use NMR to probe the histone H2A and H2B tails in the unwrapped nucleosome. Consequently, both the H2A and H2B N-tails on the pAID-proximal side bind to pAID with robust interactions, which are important for nucleosome assembly with FACT. Furthermore, the conformations of these N-tails on the distal DNA-contact site are altered from those in the canonical nucleosome. Our findings highlight that FACT both proximally and distally regulates the conformations of the H2A and H2B N-tails in the asymmetrically unwrapped nucleosome.

[1] Graduate School of Medical Life Science, Yokohama City University, 1-7-29 Suehiro-cho, Tsurumi-ku, Yokohama 230-0045, Japan. [2] Graduate School of Integrated Sciences for Life, Hiroshima University, 1-4-4 Kagamiyama, Higashi-Hiroshima 739-8528, Japan. ✉email: nisimura@yokohama-cu.ac.jp

In eukaryotes, chromatin is hierarchically packed by arrays of nucleosomes, which are the fundamental structural units, in which 145–147-bp of DNA is wrapped around a histone octamer comprising one histone H3/H4 tetramer and two histone H2A/H2B dimers[1]. Nucleosome plays a crucial role in genome stability, storage of epigenetic information, and restriction of DNA accessibility[2–6]. Hexasome, which is a nucleosome lacking one H2A/H2B dimer[7,8], and a partially unwrapped nucleosome[4] are formed as intermediates during the nucleosome assembly and disassembly by histone chaperones[9,10], ATP-dependent chromatin remodelers[11,12], and RNA polymerase II[13,14]. Such intermediates have been observed in transcribed genes in vivo[15,16], and play important roles in altering DNA accessibility and epigenetic information[3,4,6].

Each histone in nucleosome has a structured core region with a disordered N- and C-terminal tail (N- and C-tail, respectively), protruding from the core structure. The histone tails are responsible for nucleosomal stability[17], inter-nucleosomal interaction[18], DNA unwrapping[19], nucleosome sliding[20,21], and protein accessibility[22–25]. Specific residues of the histone tails are known to be targets for post-translational modifications, which act as epigenetic markers to regulate chromatin accessibility and gene expression[5,26–30]. However, characterizing the structural ensembles of histone tails at atomic resolution is difficult because of their conformational flexibility. At present, NMR analysis is the only method available to directly measure the dynamic ensembles of disordered tails within nucleosome[31–35]. In addition, the structural dynamics of the histone core region have been characterized by methyl-based NMR spectroscopy[36–39] and solid-state NMR[31].

The histone chaperone FACT (facilitate chromatin transcription) conducts nucleosome assembly/disassembly during cell differentiation[40,41], transcription[15,42–44], DNA replication[45], DNA repair[46,47], chromatin maintenance[48–50], and virus infection[51] among other processes. We have previously documented several structural alterations that occur during the nucleosome disassembly by human FACT (hFACT), comprising SPT16 and SSRP1 subunits[9,10,24]. First, we showed that the phosphorylated acidic intrinsically disordered (pAID) segment of human SPT16 binds to the H2B N-tail, which is detached from the nucleosomal DNA by a double-strand break; subsequently, the adjacent Mid domain of human SPT16 correctly interacts with the H2A docking surface of the H3/H4 tetramer, thereby displacing the H2A/H2B dimer from nucleosome, through steric collisions to form hexasome[9]. In agreement with that finding, human SPT16 has been shown to displace H2A-H2B dimers from nucleosome to unwrap the nucleosomal DNA together with DNA stretching[52]. Second, we revealed the cryo-electron microscopy (cryo-EM) structure of a partially unwrapped nucleosome complexed with hFACT, in which 112-bp DNA and pAID are asymmetrically wrapped around the histone octamer (Fig. 1a, 112-bp DNA/pAID nucleosome)[10]. This structure highlights that pAID of hFACT retains the nucleosome core structure instead of DNA. Similar interactions between FACT and unwrapped nucleosomes have been recently observed[53–55]; however, the histone tails are not visualized in those cryo-EM structures. For the 112-bp DNA/pAID nucleosome, our previous NMR study clarified that the H3 N-tails, which are invisible in the cryo-EM structure, adopt two distinct conformations reflecting their asymmetric locations; a conformation of contact to DNA, as in the canonical nucleosome where the H3 N-tail is buried in two DNA gyres (DNA side); and a conformation of reduced contact to DNA and pAID (pAID side)[24]. We further showed that acetylation of H3K14 by Gcn5 was much faster on the pAID side than on the DNA side or in the canonical nucleosome, highlighting that the pAID side H3 N-tail adopts the conformation that is more accessible to Gcn5 through partial replacement of DNA with pAID.

Similar to the H3 N-tails, the H2A N- and C-tails and the H2B N-tail are invisible in the cryo-EM structure of the 112-bp DNA/pAID nucleosome, and are also located in asymmetric environments (Fig. 1a)[10]. The H2B N-tail and the H2A C-tail stick out between two DNA gyres on the DNA side, whereas those on the pAID side are sandwiched between the pAID segment and one DNA strand (Fig. 1b, c). Regarding the two H2A N-tails, one is close to pAID, while the other is close to DNA (Fig. 1b). However, our recent NMR analysis and molecular dynamics (MD) simulations revealed that the H2A N-tail and the C-terminal half of the H2B N-tail adopt two different conformations of contact and reduced contact to DNA even in the symmetric nucleosome, and the conformations of the H2A and H2B N-tails are correlated with each other[33]. Thus, the H2A and H2B tails in the asymmetrically unwrapped nucleosome are expected to adopt multiple conformations with much more complicated modes as compared with the H3 N-tail.

Here, we have clarified the dynamic structure of H2A and H2B tails in the 112-bp DNA/pAID nucleosome and 112-bp hexasome, by using NMR. The multiple conformations of H2A and H2B tails in the 112-bp DNA/pAID nucleosome can be clearly distinguished as the proximal pAID-side and the distal DNA side. The H2A and H2B N-tails on the pAID side more frequently contact pAID relative to their contact with DNA in the canonical nucleosome. The H2B N-tail on the DNA side has significantly increased contacts with DNA, whereas the DNA side H2A N-tail adopts mainly a conformation of reduced contact to DNA. Thus, within nucleosome, hFACT seems to modulate chromatin signaling and accessibility of the H2A and H2B N-tails on both the proximal and distal sides in contrast to its modulation of only the proximal H3 N-tail.

## Results

**NMR signals of H2A and H2B tails in the 112-bp DNA/pAID nucleosome.** To investigate the chemical environments around histone H2A and H2B tails in the asymmetrically unwrapped nucleosome with hFACT, we measured the 2D $^1$H-$^{15}$N heteronuclear single quantum coherence (HSQC) spectrum of the 112-bp DNA/pAID nucleosome, reconstituted by a salt dialysis method[24], with $^2$H/$^{15}$N/$^{13}$C-labeled H2A and H2B in 20 mM HEPES-NaOH, pH 7.0 (Fig. 1d). Consistent with a previous study of the canonical nucleosome[33], NMR signals from residues within the structural core were unobservable. The signal assignments confirmed that the observed peaks derived from residues Ser1–Lys9 and Thr120–Lys129 of the histone H2A tail with two additional tag residues, Met0 and Gly-1, and residues Glu2, Ala4–Ala7, Ala9, Lys11–Lys27, and Lys125 of the histone H2B N-tail with two additional tag residues, Met0 and Gly-1 (Fig. 1d, e). Notably, we observed doublet signals for Gly2, Arg3, Lys5, Gln6, Gly8, and Ala126–Gly128 of the H2A tail and Ser6, Lys16–Vla18, and Lys23–Gly26 of the H2B tail (Supplementary Fig. 1). In addition, we observed triplet signals for Ser1, Gly4, and Gly7 of the H2A tail.

**Signal identification of pAID and DNA side tails in the 112-bp DNA/pAID nucleosome.** To determine which are the pAID side signals of the H2A and H2B tails, we titrated 33-bp DNA into the 112-bp DNA /pAID nucleosome incorporating labeled H2A and H2B at molar ratios of 0.5:1, 1:1, and 2:1 (Supplementary Fig. 2a–d). The signals of Ser1, Arg3–Gly8, and Ala126–Gly128 of H2A, and Lys16–Gly26 of H2B were significantly changed upon the addition of an equivalent amount of DNA (Supplementary Fig. 2c), and remained almost the same up to the two-fold

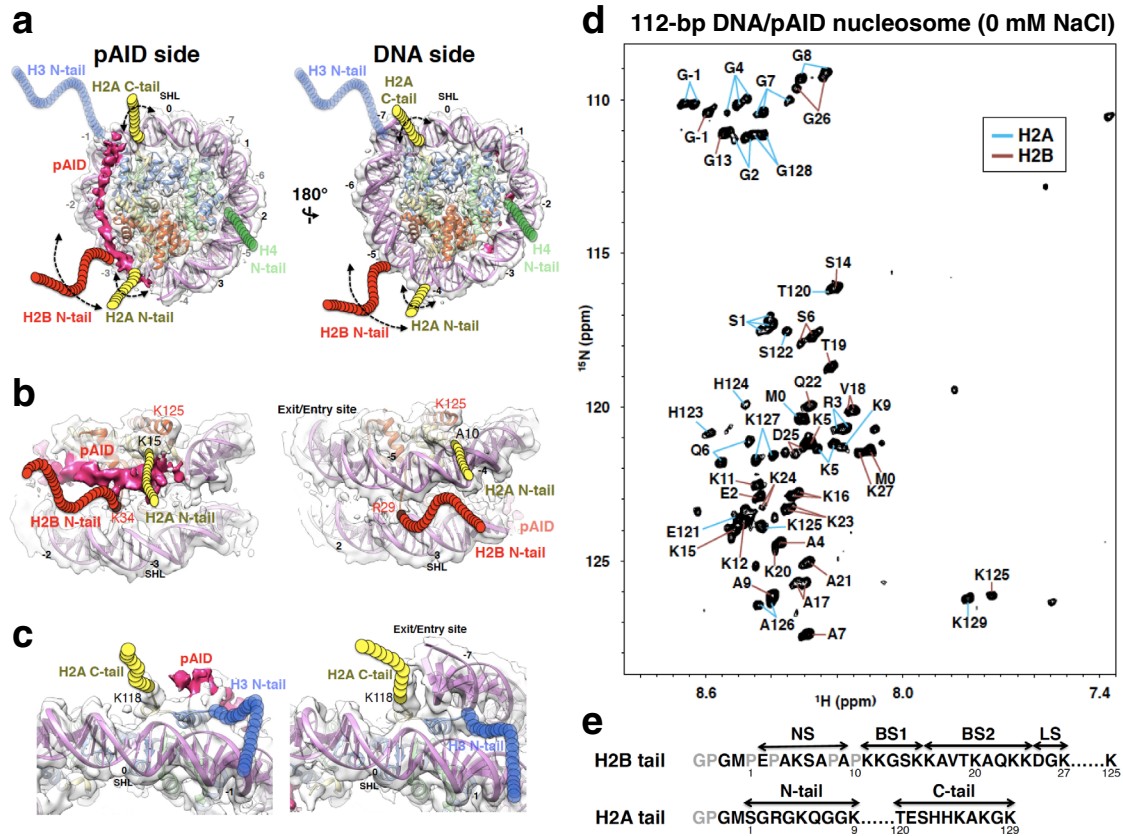

**Fig. 1 2D $^1$H-$^{15}$N HSQC spectrum of the H2A and H2B tails in the 112-bp DNA/pAID nucleosome. a–c** Cryo-EM structure of the 112-bp DNA/pAID nucleosome. Six views of the EM density map (EMD-9639) are superimposed on the nucleosome structure (PDB ID: 2CV5) lacking the 33-bp DNA. Visualization of the EM maps and fitting of the nucleosome structures in the maps were performed using UCSF Chimera[67]. pAID is shown as deep pink density. H2A, H2B, H3, H4, and DNA are colored yellow, red, blue, green, and orchid, respectively. Colored circular chains denote individual histone tails, which are not observed in the EM structure. Superhelix locations (SHL), representing the number of double turns from the dyad axis of the canonical nucleosome (0), are indicated. **d** Backbone resonance assignments of the 112-bp DNA/pAID nucleosome in the 2D $^1$H-$^{15}$N HSQC spectrum recorded in 20 mM HEPES-NaOH, pH 7.0, 10% $D_2O$ on a 950-MHz spectrometer. Blue and orange lines indicate residues of H2A and H2B, respectively. **e** Amino acid sequences of the H2A and H2B tails. Assigned and unassigned residues in NMR are colored black and gray, respectively.

addition of DNA (Supplementary Fig. 2d). This result suggests that the saturation point is more or less reached on equivalent addition, and the excess 33-bp DNA present after the two-fold addition of DNA hardly affects the chemical shifts. Addition of 33-bp DNA to the 112-bp DNA/pAID nucleosome has been shown to lead to the replacement of pAID with DNA, resulting in a double-strand break nucleosome wrapped by 112-bp and 33-bp DNA (112/33-bp nucleosome) (Supplementary Fig. 2a). Upon the addition of DNA, the asymmetric conformations of histone H2A, H2B, and H3 tails in the 112-bp DNA/pAID nucleosome should converge into symmetric conformations similar to those in the canonical nucleosome. In fact, we previously showed that, of two signals observed for each H3 N-tail residue, one corresponding to the pAID side almost disappeared or was impaired, whereas the other corresponding to the DNA side remained in the same position, on titration of 33-bp DNA into the 112-bp DNA/pAID nucleosome[24].

Here, regarding the H2A N-tail, the two signals of Ser1, the highest-field signal of Gly4, Lys5, Gln6, Gly7, and Gly8, and one signal of Lys9 disappeared when up to a two-fold excess of 33-bp DNA was titrated into the 112-bp DNA/pAID nucleosome (Fig. 2a); thus, these signals were ascribed to the pAID side signals, and the remaining signals were assigned to the DNA side. Hereafter, signals corresponding to the pAID and DNA side of each amino acid are designated by one-letter representation with subscript p and d, respectively (Fig. 2a, b); similarly, high- and

low- field signals on each side are designated as subscript h and l, respectively. Thus, the signals from the pAID side H2A N-tail are denoted $S1_{ph}$, $S1_{pl}$, $G4_p$, $K5_p$, $Q6_p$, $G7_p$, $G8_p$, and $K9_p$, while the DNA side signals are designated $S1_d$, $G4_{dh}$, $G4_{dl}$, $K5_d$, $Q6_d$, $G7_{dh}$, $G7_{dl}$, and $G8_d$ (Fig. 2a). Of note, the doublet signals for Gly2 and Arg3 ($G2_{h, l}$ and $R3_{h, l}$) showed no significant changes after 33-bp DNA addition (Fig. 2a), indicating that the conformations of Gly2 and Arg3 were hardly affected by the replacement of pAID with DNA.

Regarding the H2A C-tail, both of the doublet signals of $A126_{h, l}$, $K127_{h, l}$, and $G128_{h, l}$ slightly shifted to the higher field after the 33-bp DNA addition (Fig. 2a), suggesting that these residues maintained two conformations, both of which were slightly altered by the replacement of pAID with DNA. In addition, the signals of Thr120–Lys125 and Lys129 remained in similar positions after the 33-bp DNA addition (Fig. 2a). Therefore, all residues in the H2A C-tail seem to adopt similar conformations on the pAID and DNA sides.

Surprisingly, for most residues in the H2B N-tails, we could not identify apparent pAID side signals, except for the higher field signal of Lys16 ($K16_p$), and the lower field signals of Asp25 ($D25_p$) and Gly26 ($G26_p$) in the N-tail, and the signal of Lys125 ($K125_p$) in the C-tail (Fig. 2b); these few pAID side signals disappeared upon the 33-bp DNA addition, while their counterpart DNA side signals remained ($K16_d$ and $D25_d$) or shifted ($G26_d$) in position.

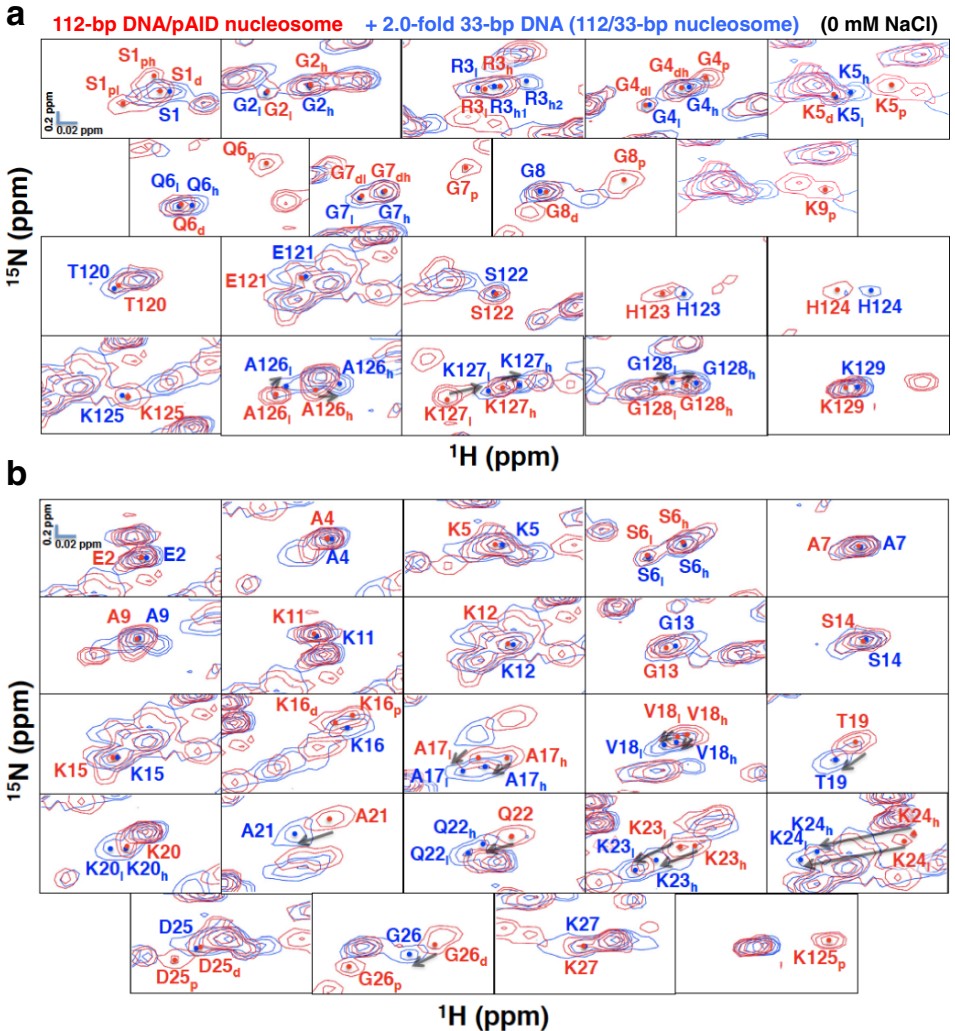

**Fig. 2 Comparison of extended HSQC spectra of the H2A and H2B tails between the 112-bp DNA/pAID nucleosome alone (red) and with the two-fold addition of 33-bp DNA (blue). a, b** NMR signals are divided into panels by amino acid residue of H2A (**a**) and H2B (**b**). Signal assignments in the 112-bp DNA/pAID nucleosome alone and with the two-fold addition of DNA at 0 mM NaCl are labeled in red and blue, respectively. Two-fold addition of 33-bp DNA to the 112-bp DNA/pAID nucleosome leads to a double-strand break nucleosome wrapped by 112-bp and 33-bp DNA (112/33-bp nucleosome), as shown in Supplementary Fig. 2a. Filled circles indicate each signal center. The signal centers were identified from peaks in the slice representations of signals in Supplementary Fig. 1. pAID and DNA side signals are designated by subscript p and d, respectively. High- and low-field components of the signals are designated by subscript h and l, respectively. Arrows represent chemical shift changes upon the addition of 33-bp DNA.

According to previous NMR dynamics reported for the H2B N-tail[33], we classified the H2B N-tail into four segments (Fig. 1e): an N-terminal segment from Glu2 to Ala9 (NS); two basic segments from Lys11 to Lys15 (BS1), and from Lys16 to Lys24 (BS2); and a linker segment from Asp25 to Lys27 (LS), which connects BS2 to the nucleosome core. The NS and BS1 regions and the last residue of LS (Lys27) in the H2B N-tail showed no significant signal changes after the 33-bp DNA addition (Fig. 2b). Thus, the NS and BS1 regions in the H2B N-tail seem to fluctuate similarly on both the pAID and DNA sides, resulting in no significant signal differences between two sides. On the other hand, the BS2 region of the H2B N-tail showed a marked low-field shift of singlet (T19, A21, and Q22$_h$) and doublet (A17$_{h, l}$, V18$_{h, l}$, K23$_{h, l}$, and K24$_{h, l}$) signals, while the K20$_l$ and Q22$_l$ signals newly appeared upon the addition of 33-bp DNA (Fig. 2b). As discussed below, it is possible that signals for the pAID side H2B N-tail are hardly observed owing to the robust interaction with pAID, while those for the DNA side H2B N-tail are observed only in the 112-bp DNA/pAID nucleosome.

**Establishment of the pAID and DNA side signals of the H2B BS2 region in the 112-bp DNA/pAID nucleosome**. To establish further the DNA side signals of the H2B N-tail in the 112-bp DNA/pAID nucleosome, we measured the HSQC spectra of the 112-bp hexasome, in which one H2A/H2B dimer is absent (Supplementary Fig. 3a, b)[10], reconstituted with 112-bp DNA, an H3/H4 tetramer, and one equivalent of H2A/H2B dimer incorporating $^2$H/$^{13}$C/$^{15}$N-labeled H2A and H2B. Owing to the absence of the pAID side (Supplementary Fig. 3a), the conformation of the H2A and H2B tails in the 112-bp hexasome should reflect the DNA side conformation in the 112-bp DNA/pAID nucleosome. In fact, the DNA side signals of the H2B N-tail in the 112-bp DNA/pAID nucleosome (D25$_d$ and G26$_d$ in LS) roughly corresponded to those in the 112-bp hexasome (D25 and G26 in LS) (Fig. 3a). In addition, most unclassified signals in the H2B N-tail of the 112-bp DNA/pAID nucleosome (E2, A4, K5, S6$_{h, l}$, and A7 in NS; A9 and K11–K15 in BS1; A17$_h$, V18$_{h, l}$, T19, K20, A21, Q22, K23$_h$, and K24$_l$, in BS2; and K27 in LS) nearly corresponded to their counterpart signals in the 112-bp hexasome (E2, A4, K5, S6$_{h, l}$, and A7 in NS; A9 and K11–K15 in

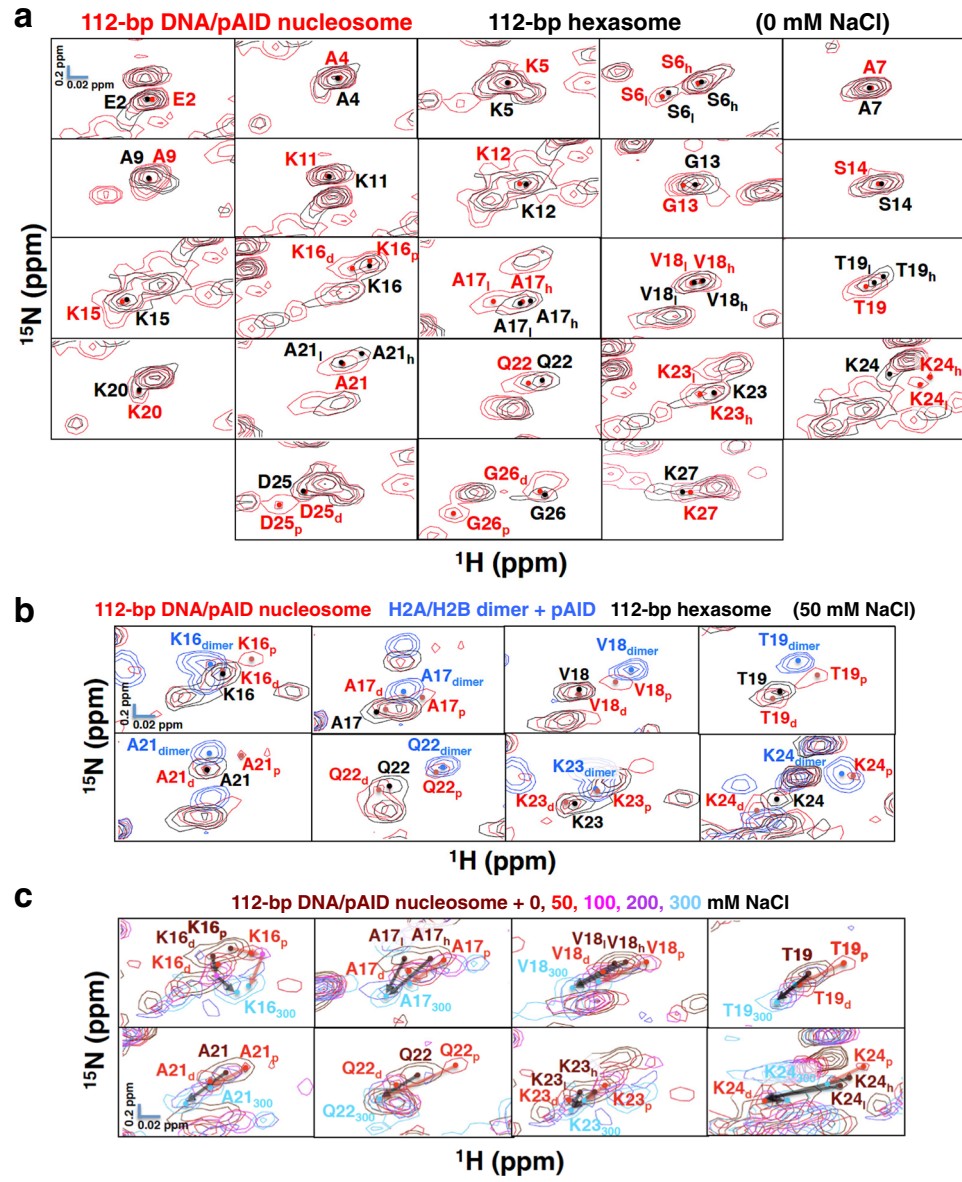

**Fig. 3 Establishment of the pAID and DNA sides of the H2B N-tail in the 112-bp DNA/pAID nucleosome. a** Comparison of extended HSQC spectra of the H2B N-tail between the 112-bp DNA/pAID nucleosome (red) and the 112-bp hexasome (black) at 0 mM NaCl. NMR signals are divided into panels by amino acid residue of H2B. Signal assignments in the 112-bp DNA/pAID nucleosome and 112-bp hexasome at 0 mM NaCl are labeled in red and black, respectively. Filled circles indicate each signal center. pAID and DNA side signals are designated by subscript p and d, respectively. High- and low-field components of the signals are designated by subscript h and l, respectively. **b** Expanded signal comparison of the H2B BS2 region among the 112-bp DNA/ pAID nucleosome (red), 112-bp hexasome (black), and H2A/H2B dimer upon the addition of an equivalent amount of pAID (light blue) at 50 mM NaCl. Residues corresponding to signals of the H2A/H2B dimer upon the addition of pAID are designated by the subscript "dimer". **c** Ionic strength dependence of selected signals of the H2B BS2 region in the 112-bp DNA/pAID nucleosome upon NaCl titration at 0 M (brown), 50 mM (red), 100 mM (pink), 200 mM (purple), and 300 mM (cyan). NMR signals are divided into panels by amino acid residue. Residues are labeled in the color corresponding to the NaCl condition. Residues of the signal at 300 mM NaCl are designated by subscript 300. Red and black arrows indicate the chemical shift changes of the pAID and DNA side signals, respectively, between 0 or 50 mM and 300 mM NaCl.

BS1; A17l, V18h, l, T19l, K20, A21l, Q22, K23, and K24 in BS2; and K27 in LS) (Fig. 3a). For the present, it seems likely that the unclassified signals of the NS (E2, A4, K5, S6h, l, and A7), BS1 (A9 and K11–K15), BS2 (A17h, V18h, l, T19, K20, A21, Q22, K23h, and K24l) and LS (K27) regions of the H2B N-tail correspond to the DNA side in the 112-bp DNA/pAID nucleosome.

To compare the binding manner of each H2B N-tail residue between pAID and DNA, we measured the HSQC spectrum of a $^{13}$C, $^{15}$N-peptide comprising H2B N-tail residues Pro1–Tyr37 with four additional tag residues, Gly-3, Pro-2, Gly-1, and Met0,

after the addition of an equivalent amount of 33-bp 601 DNA or pAID protein in 20 mM HEPES-NaOH, pH 7.0 (Supplementary Fig. 4a). The signals of the H2B peptides bound to DNA and pAID were compared by superposition of the corresponding HSQC spectra. Signals for Glu2, Ala4–Ala7, and Ala9 in NS and Lys11–Lys15 in BS1 of the pAID-bound H2B peptide corresponded or were slightly shifted relative to those of the DNA-bound H2B peptide (Supplementary Fig. 4a). In contrast, signals for Lys16–Lys27 in BS2 and LS of the pAID-bound H2B peptide were greatly shifted to higher field relative to those of the DNA-

bound H2B peptide (Supplementary Fig. 4a). The chemical shift changes between the DNA- and pAID-bound H2B peptides for Lys16–Lys24 in BS2 were significantly larger than the corresponding changes in the other NS and BS1 regions (Supplementary Fig. 4b). These results suggest that the BS2 region in the H2B N-tail interacts with pAID than with DNA in a different manner, whereas the NS and BS1 regions interact dynamically with DNA and pAID in a similar manner. Presumably owing to the strong interaction with pAID at 0 mM NaCl, the pAID side signals were hardly observed in the BS2 region of the 112-bp DNA/pAID nucleosome (Fig. 2b). Namely, all signals observed for the H2B BS2 region in the 112-bp DNA/pAID nucleosome at 0 mM NaCl ($A17_{h, l}$, $V18_{h, l}$, T19, K20, A21, Q22, $K23_{h, l}$, and $K24_{h, l}$), except for Lys16, are assignable to the DNA side signals.

To confirm the idea that the pAID side signals are not observed at 0 mM NaCl because of the tight interaction of BS2 with pAID, we measured the chemical shift perturbations upon titration of NaCl at 50, 100, 200, and 300 mM into the 112-bp DNA/pAID nucleosome (Supplementary Fig. 5a–d) and 112-bp hexasome (Supplementary Fig. 6a–d). At high salt concentrations, electrostatic interactions between the histone tails and DNA or pAID are weakened, and thus the populations of their contact states with DNA or pAID will be reduced.

Importantly, regarding the H2B BS2 region, NaCl titration into the 112-bp DNA/pAID nucleosome led to the appearance of additional higher field signals ($A17_p$, $V18_p$, $T19_p$, $Q22_p$, $K23_p$, and $K24_p$) except for Lys20 at 50 mM NaCl (Fig. 3b and Supplementary Fig. 5a). These additional signals ($A17_p$, $V18_p$, $T19_p$, $Q22_p$, $K23_p$, and $K24_p$) roughly corresponded to those of the pAID-bound H2A/H2B dimer incorporating $^2H/^{13}C/^{15}N$-labeled H2B under the same conditions ($A17_{dimer}$, $V18_{dimer}$, $T19_{dimer}$, $Q22_{dimer}$, $K23_{dimer}$, and $K24_{dimer}$), and were not observed in the 112-bp hexasome without pAID (Fig. 3b). In addition, other original signals in BS2 ($K16_d$, $A17_d$, $V18_d$, $T19_d$, $A21_d$, $Q22_d$, $K23_d$, and $K24_d$) closely corresponded to those of the 112-bp hexasome under the same condition (Fig. 3b). Thus, we confirmed that the additional signals that appeared in the BS2 region ($K16_p$, $A17_p$, $V18_p$, $T19_p$, $A21_p$, $Q22_p$, $K23_p$, and $K24_p$) are ascribed to the pAID side, while the other signals ($K16_d$, $A17_d$, $V18_d$, $T19_d$, $A21_d$, $Q22_d$, $K23_d$, and $K24_d$) are assigned to the DNA side (Fig. 3b). The pAID side signal for Lys20 in BS2 was not observed at 50 mM NaCl, owing to severe overlap with the higher-filed Ala4 signal (Supplementary Fig. 5a). Eventually, the pAID side signals ($K16_p$, $A17_p$, $V18_p$, $T19_p$, $A21_p$, $Q22_p$, $K23_p$, and $K24_p$) and the DNA side signals ($K16_d$, $A17_d$, $V18_d$, $T19_d$, $A21_d$, $Q22_d$, $K23_d$, and $K24_d$) at 50 mM NaCl were merged or approached each other at the lower field position at 300 mM NaCl (Fig. 3c), indicating that the asymmetric conformations of the H2B BS2 region converge into one conformation that is dissociated from pAID and/or DNA at higher NaCl concentration.

The signal behavior of Lys16 in BS2 of the H2B N-tail in the 112-bp DNA/pAID nucleosome was different from those of the other BS2 residues. For instance, the pAID side signal of Lys16 was observed at 0 mM NaCl (Fig. 2b), while the DNA side signal of Lys16 remained in a similar position upon the DNA addition, similar to the behavior of the neighboring BS1 region (Fig. 2b). It is likely that the different behavior of Lys16 in BS2 may reflect its position at the junction between BS1 and BS2.

### Conformation of the H2A N-tails in the 112-bp DNA/pAID nucleosome.
To further investigate the conformation of the H2A N-tails, we measured the HSQC spectra of the 145-bp nucleosome incorporating $^2H/^{13}C/^{15}N$-labeled H2A and H2B (Supplementary Fig. 3c), as well as the chemical shift perturbations upon

titration of NaCl at 50, 100, 200, and 300 mM into the 145-bp nucleosome (Supplementary Fig. 7a–d).

Regarding the H2A N-tail in the 112-bp DNA/pAID nucleosome, the pAID side signals ($S1_{ph}$, $G4_p$, $K5_p$, $Q6_p$, $G7_p$, and $G8_p$) and the DNA side signals ($S1_d$, $G4_{dh, dl}$, $K5_d$, $Q6_d$, $G7_{dh, dl}$, and $G8_d$) at 0 mM NaCl were converged or approached each other at 300 mM NaCl (Fig. 4a). In addition, a higher field signal for Arg3 ($R3_p$) additionally appeared in the 112-bp DNA/pAID nucleosome at 50 mM NaCl, while a corresponding signal was not observed in the 112-bp hexasome without pAID under the same condition (Supplementary Fig. 8a). Other signals ($R3_{dh}$ and $R3_{dl}$) in the 112-bp DNA/pAID nucleosome corresponded to those in the 112-bp hexasome under the same condition. Therefore, the additional signal ($R3_p$) is ascribed to the pAID side conformation of H2A, and the others ($R3_{dh}$ and $R3_{dl}$) are due to the DNA side. The pAID side $R3_p$ signal and the DNA side $R3_{dh, dl}$ signals at 50 mM NaCl were also converged at 300 mM NaCl (Fig. 4a). A pAID side signal was not observed for Gly2 under any condition, owing to severe overlap with the higher field Gly128 signal (Fig. 4a).

Even under the physiological salt condition (100 mM NaCl), the pAID side signals ($S1_p$, $R3_p$, $K5_p$, $Q6_p$, $G7_p$, and $G8_p$) were observed in the 112-bp DNA/pAID nucleosome, but not in the 145-bp nucleosome or the 112-bp hexasome (Fig. 4b, c and Supplementary Table 1). The chemical shift changes at $S1_p$, $R3_p$, $K5_p$, $Q6_p$, $G7_p$, and $G8_p$ on the pAID side between 100 mM and 300 mM NaCl were significantly larger than the corresponding changes on the DNA side ($S1_d$, $R3_{dh, dl}$, $K5_d$, $Q6_d$, $G7_d$, and $G8_d$) (Fig. 4d), indicating that, under the physiological salt condition, the pAID side H2A N-tail more frequently contacts pAID as compared with the DNA side residues contacting DNA.

At 100 mM NaCl, we compared the DNA side signals for the H2A N-tail among the 112-bp DNA/pAID nucleosome, 112-bp hexasome, and 145-bp nucleosome (Fig. 4b, c and Supplementary Table 1). Doublet signals of the H2A N-tail were observed in the 145-bp nucleosome: one signal ($G2_l$, $G4_l$, $K5_h$, $Q6_l$, and $G7_l$) and the other ($G2_h$, $G4_h$, $K5_l$, $Q6_h$, and $G7_h$) reflecting the DNA contact and reduced-contact conformations of the H2A N-tail, respectively[33] (Fig. 4b and Supplementary Table 1). In contrast, the DNA side signals of the H2A N-tail in the 112-bp DNA/pAID nucleosome at 100 mM NaCl were observed as a singlet signal ($G2$, $G4$, $K5_d$, $Q6_d$, $G7_d$), corresponding to the reduced-contact conformation signals in the 145-bp nucleosome ($G2_h$, $G4_h$, $K5_l$, $Q6_h$, and $G7_h$)[33] (Fig. 4b and Supplementary Table 1). Interestingly, these signals also corresponded to the H2A N-tail signals in the 112-bp hexasome at 100 mM NaCl (S1–S8) (Fig. 4c and Supplementary Table 1). These results suggest that, under the physiological salt condition, the H2A N-tail on the DNA side of the 112-bp DNA/pAID nucleosome, as well as in the hexasome, mainly adopts the reduced-contact conformation observed in the 145-bp nucleosome.

To further investigate the conformation of the H2A N-tail on the pAID side in the 112-bp DNA/pAID nucleosome, we compared these pAID side signals with the corresponding signals of the H2A/H2B dimer incorporating $^2H/^{13}C/^{15}N$-labeled H2A after the addition of an equivalent amount of pAID at 0 mM NaCl. The chemical shifts of the pAID side signals of $S1_{ph}$, $G4_p$, $K5_p$, $Q6_p$, $G7_p$, $G8_p$, and $K9_p$ differed significantly from the corresponding chemical shifts of both the pAID-bound H2A/H2B dimer ($S1_{dimer}$, $G4_{dimer}$, $K5_{dimer}$, $Q6_{dimer}$, $G7_{dimer}$, $G8_{dimer}$, and $K9_{dimer}$) and the DNA side signals ($S1_d$, $G4_{dh}$, $G4_{dl}$, $K5_d$, $Q6_d$, $G7_{dh}$, $G7_{dl}$, and $G8_d$) (Supplementary Fig. 8b). These results suggest that the pAID-bound residues of the H2A N-tail within the 112-bp DNA/pAID nucleosome adopt an entirely different conformation from their counterparts either in the pAID-bound

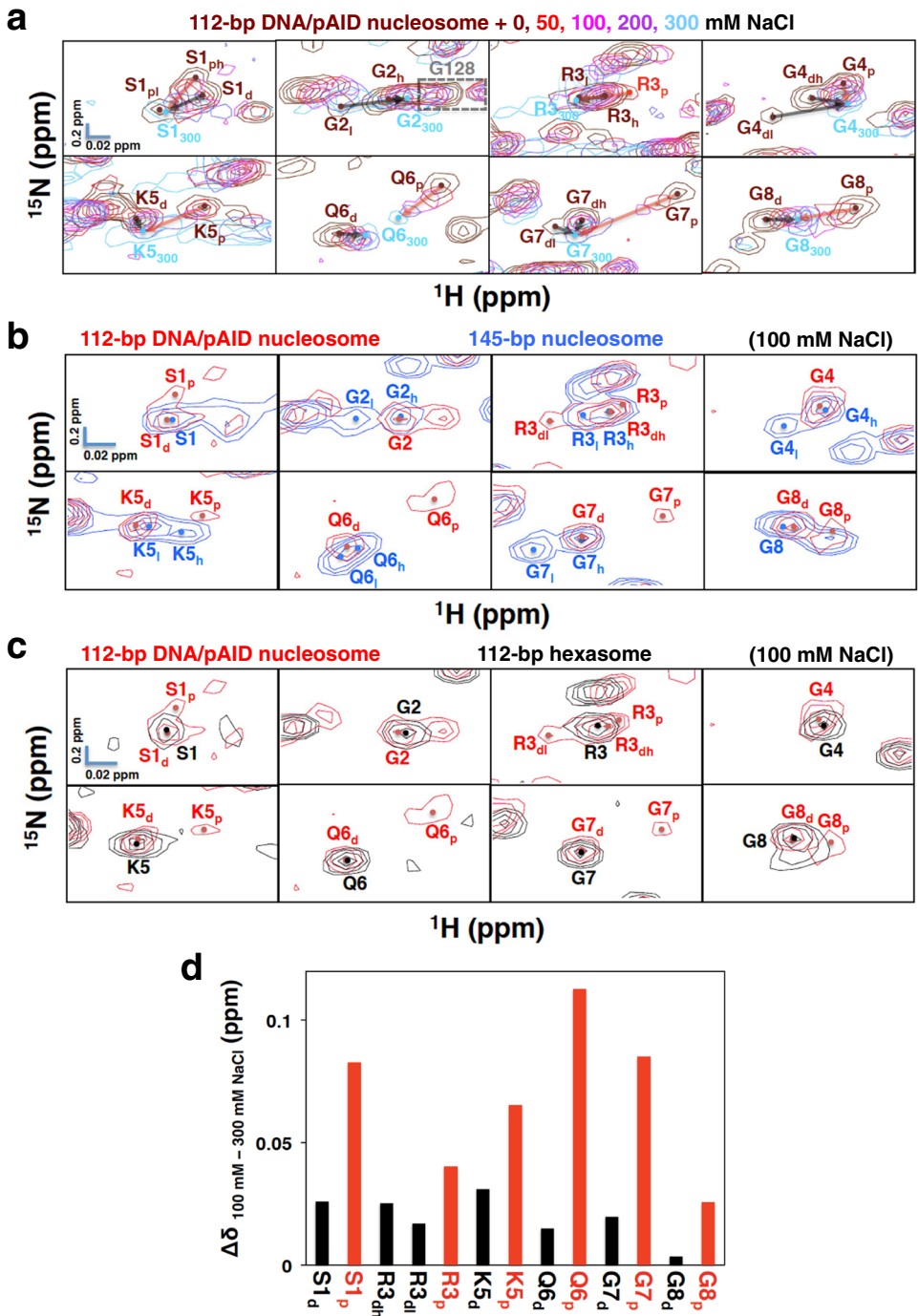

**Fig. 4 Conformational comparison of the H2A N-tails under the physiological salt condition. a** Ionic strength dependence of signals of the H2A N-tail in the 112-bp DNA/pAID nucleosome upon NaCl titration at 0 M (brown), 50 mM (red), 100 mM (pink), 200 mM (purple), and 300 mM (cyan). NMR signals are divided into panels by amino acid residue. Residues are labeled in the color corresponding to the NaCl condition. Residues of the signal at 300 mM NaCl are designated by subscript 300. Red and black arrows indicate the chemical shift changes of the pAID and DNA side signals, respectively, between 0 or 50 mM and 300 mM NaCl. pAID and DNA side signals are designated by subscript p and d, respectively. High- and low-field components of the signals are designated by subscript h and l, respectively. **b, c** Expanded signal comparison between the 112-bp DNA/pAID nucleosome (red) and the 145-bp nucleosome (blue) (**b**) or the 112-bp hexasome (black) (**c**) at 100 mM NaCl. NMR signals are divided into panels by amino acid residue (Ser1–Gly8 of H2A). Signal assignments in the 112-bp DNA/pAID nucleosome, 145-bp nucleosome, and 112-bp hexasome are labeled in red, blue, and black, respectively. Filled circles indicate each signal center. **d** Histogram showing chemical shift differences between 100 mM and 300 mM NaCl of the pAID and DNA side signals in the H2A N-tail of the 112-bp DNA/ pAID nucleosome. Chemical shift differences are plotted for pAID side (red) and DNA side (black) residues in the H2A N-tail.

H2A/H2B dimer or on the DNA side. In the 112-bp DNA/pAID nucleosome at 0 mM NaCl, however, the chemical shift indices[56,57] of Gly4–Gly8 in the H2A N-tail were very similar between the pAID and DNA sides, showing no structural

elements (Supplementary Fig. 8c). This suggests that the H2A N-tails on both sides adopt a similar disordered structure, although they bind respectively to pAID and DNA with different binding modes within nucleosome.

**Conformation of the H2A C-tails in the 112-bp DNA/pAID nucleosome**. According to the EM structure of the 112-bp DNA/pAID nucleosome[10], the two H2A C-tails are also located in asymmetric positions (Fig. 1c). At 0 mM NaCl, however, the signals of T120–K125, A126$_h$, K127$_h$, G128$_h$, and K129 of the H2A C-tail in the 112-bp DNA/pAID nucleosome roughly corresponded to the signals of T120–K129 in the 145-bp nucleosome and 112-bp hexasome (Fig. 1d and Supplementary Figs. 3b, c). Furthermore, at 100 mM NaCl, the signals observed for the H2A C-tail in the 112-bp DNA/pAID nucleosome (E121, S122, K125, A126, K127$_h$, G128$_h$, K129$_h$) were equivalent to the corresponding signals in the 145-bp nucleosome and 112-bp hexasome (E121, S122, K125–K129) (Supplementary Figs. 5b, 6b, and 7b and Supplementary Table 1). These results suggest that the H2A C-tail fundamentally adopts a symmetrical conformation regardless of its asymmetric locations in the complexes, probably due to a freely fluctuating conformation or dynamic interaction with a symmetric DNA strand around the dyad axis.

The irreconcilable low-field signals (A126$_l$, K127$_l$, and G128$_l$) in the 112-bp DNA/pAID nucleosome and 112/33-bp nucleosome at 0 mM NaCl roughly corresponded to those in the 112-bp hexasome after the addition of an equivalent amount of pAID (Supplementary Figs. 3d and 8d). However, the other residues in the 112-bp hexasome were hardly changed upon the addition of pAID (Supplementary Fig. 3b, d). This suggests that pAID partially interacts with Ala126, Lys127, and Gly128 of the H2A C-tail in the 112-bp DNA/pAID nucleosome. Probably, in the 112/33-bp nucleosome, the signals may derive from the interaction of these residues with free pAID that is displaced upon the addition of DNA.

**Conformation of the H2B N-tails in the 112-bp DNA/pAID nucleosome**. Consistent with this result for the H2B peptide (Supplementary Fig. 4), the signals for the NS and BS1 regions (Glu2–Lys15) of the H2B N-tail in the 112-bp DNA/pAID nucleosome could not be assigned to the pAID or DNA side, and hardly differed among the 112-bp DNA/pAID nucleosome, 145-bp nucleosome, and 112-bp hexasome at any concentration of NaCl (Supplementary Figs. 5a–d, 6a–d, and 7a–d and Supplementary Table 2). These results suggest that the conformation of the NS and BS1 regions of the H2B N-tail is similar on the pAID and DNA sides, probably due to similar dynamic interactions with DNA and pAID.

Regarding the H2B BS2 region, the pAID side signals were observed in the 112-bp DNA/pAID nucleosome even under the physiological salt condition (K16$_p$, A17$_p$, V18$_p$, A21$_p$, and K23$_p$), but not in the 145-bp nucleosome or the 112-bp hexasome (Fig. 5a, b and Supplementary Table 2). The chemical shift changes of the BS2 residues on the pAID side (K16$_p$, A17$_p$, V18$_p$, A21$_p$, and K23$_p$) between 100 and 300 mM NaCl were slightly larger than the corresponding changes on the DNA side (K16$_d$, A17$_d$, V18$_d$, A21$_d$, and K23$_{dh, dl}$) (Fig. 5c), indicating that the BS2 region in the H2B N-tail has increased populations of the contact states with pAID relative to the contact states with DNA under the physiological salt condition.

Next, we compared the DNA side signals for the H2B BS2 region among the 112-bp DNA/pAID nucleosome, 112-bp hexasome, and 145-bp nucleosome at 100 mM NaCl (Fig. 5a, b and Supplementary Table 2). Doublet signals were observed for BS2 in the 145-bp nucleosome (A17$_{h, l}$, V18$_{h, l}$, T19$_{h, l}$, A21$_{h, l}$, Q22$_{h, l}$, K23$_{h, l}$, and K24$_{h, l}$), reflecting the contact and reduced-contact conformations of the BS2 region to DNA[33] (Fig. 5a and Supplementary Table 2). In contrast, the DNA side signals of BS2 in the 112-bp DNA/pAID nucleosome at 100 mM NaCl were observed as a singlet signal (K16$_d$, A17$_d$, V18$_d$, T19, K20, A21$_d$,

Q22, and K24), corresponding to the higher field signals in the 145-bp nucleosome (K16$_h$, A17$_h$, V18$_h$, T19$_h$, K20, A21$_h$, Q22$_h$, and K24$_h$), except for doublet signals at Lys23 (Fig. 5a and Supplementary Table 2). These DNA side signals also corresponded to the signals in the 112-bp hexasome at 100 mM NaCl (K16 and V18–K24), except for Ala17 (Fig. 5b and Supplementary Table 2). In the 145-bp nucleosome, the signals of A17$_h$, V18$_h$, T19$_h$, and Q22$_h$ corresponded to the DNA contact conformation, but not the reduced-contact conformation[33]. Therefore, the H2B BS2 region on the DNA side in the 112-bp DNA/pAID nucleosome, as well as in the hexasome, adopts the DNA contact conformation in the 145-bp nucleosome under the physiological salt condition.

At 0 mM NaCl, the H2B BS2 region on the DNA side in the 112-bp DNA/pAID nucleosome more clearly adopted the DNA contact conformation. In the chemical shift changes between each of 112-bp DNA/pAID nucleosome, 112-bp hexasome, and 145-bp nucleosome at 0 mM NaCl and the 112-bp DNA/pAID nucleosome at 300 mM NaCl, larger differences were observed for the BS2 region of the 112-bp hexasome and 112-bp DNA/pAID nucleosome, while the 145-bp nucleosome showed the smallest differences (Supplementary Fig. 9). Therefore, the populations of the contact states of the BS2 region to DNA are more significantly increased in the 112-bp DNA/pAID nucleosome and 112-bp hexasome, as compared with the 145-bp nucleosome.

Regarding the H2B LS region, the signals of D25, G26, and K27 at 100 mM NaCl hardly differed among the 112-bp DNA/pAID nucleosome, 145-bp nucleosome, and 112-bp hexasome (Supplementary Figs. 5b, 6b, and 7b and Supplementary Table 2).

**Role of the H2A and H2B N-tails in nucleosome assembly**. To further investigate the role of the H2A and H2B N-tails on the pAID side, we used electrophoretic mobility shift assays (EMSAs) to confirm whether nucleosome with hFACT, pAID, or DNA can assemble from hexasome and either a full-length H2A/H2B dimer or a truncated H2A/H2B mutant lacking Pro1–Lys24 of H2B (H2B_Δ24) or Ser1–Lys9 of H2A (H2A_Δ9) at 100 mM NaCl (Fig. 6a, b). The EMSAs revealed that nucleosomal complexes can be formed by combining the 112-bp hexasome with the full-length dimer and pAID, 33-bp DNA, or hFACT under the physiological salt condition (Fig. 6b, c). Note, however, that two bands corresponding to nucleosome-hFACT and nucleosome-hFACT-H2A/H2B were observed as slower bands (Fig. 6b and Supplementary Fig. 10a, b). In order to get the band intensity of the nucleosomal complexes with hFACT in Fig. 6c and Supplementary Fig. 10c, therefore, we added the band intensities of both the nucleosome-hFACT and nucleosome-hFACT-H2A/H2B complexes.

In addition, deletion of the N-tail of either H2B (H2A/H2B_Δ24 dimer) (Fig. 6b, c) or H2A (H2A_Δ9/H2B dimer) (Supplementary Fig. 10b, c) hardly affected the formation of all three complexes; however, deletion of the N-tails of both H2A and H2B (H2A_Δ9/H2B_Δ24 dimer) significantly reduced complex formation with pAID and hFACT relative to complex formation with 33-bp DNA (Fig. 6b, c). This highlights that the robust interactions of both the H2A and H2B N-tails with pAID are important for nucleosome assembly from hexasome, but are not important in the interaction with DNA. This agrees well with the NMR observation that the populations of the contact states with pAID are higher than those of the contact states with DNA in the H2A N-tail and the H2B BS2 region of the 112-bp DNA/pAID nucleosome under the physiological salt condition (Figs. 4d, 5c).

**Effect of divalent cations on the 112-bp DNA/pAID nucleosome**. To investigate the effect of divalent cations, we measured

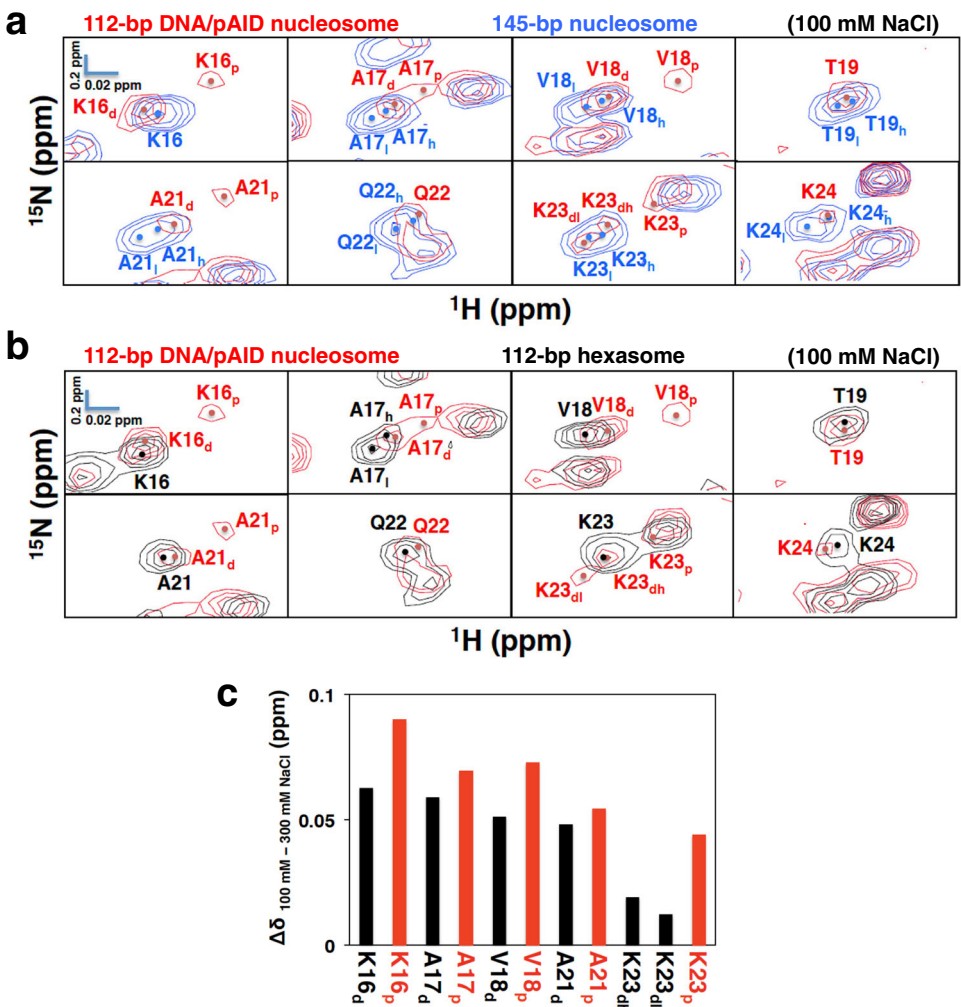

**Fig. 5 Conformational comparison of the H2B BS2 region under the physiological salt condition. a, b** Expanded signal comparison between the 112-bp DNA/pAID nucleosome (red) and the 145-bp nucleosome (blue) (**a**) or the 112-bp hexasome (black) (**b**) at 100 mM NaCl. NMR signals are divided into panels by amino acid residue (Lys16–Thr19 and Ala21–Lsy24 of H2B). Signal assignments in the 112-bp DNA/pAID nucleosome, 145-bp nucleosome, and 112-bp hexasome are labeled in red, blue, and black, respectively. Filled circles indicate each signal center. pAID and DNA side signals are designated by subscript p and d, respectively. High- and low-field components of the signals are designated by subscript h and l, respectively. **c** Histogram showing chemical shift differences between 100 mM and 300 mM NaCl of the pAID and DNA side signals in the H2B BS2 region of the 112-bp DNA/ pAID nucleosome. Chemical shift differences are plotted for pAID side (red) and DNA side (black) residues in the H2B BS2 region.

the chemical shift perturbations upon titration of $MgCl_2$ into the 112-bp DNA/pAID nucleosome and 145-bp nucleosome (Supplementary Fig. 11). Signals of the 145-bp nucleosome were observed at 0.5, 1, and 2 mM $MgCl_2$, and lost at 4 mM $MgCl_2$ owing to severe aggregation (Supplementary Fig. 11a–c). The changes in the signals of the 145-bp nucleosome observed at 2 mM $MgCl_2$ were roughly equivalent to those observed at 100 mM NaCl (Supplementary Figs. 7b, 11c). Signals of the 112-bp DNA/pAID nucleosome were observed at 0.5 mM $MgCl_2$, and lost at 1 mM $MgCl_2$ owing to severe aggregation (Supplementary Fig. 11d). The changes in the signals of the 112-bp DNA/pAID nucleosome observed at 0.5 mM $MgCl_2$ were roughly equivalent to those observed at 50 mM NaCl (Supplementary Figs. 5a, 11d), except for some signals on the pAID side in the H2A N-tail; that is, the signal shifts of $G4_p$, $K5_p$, $G7_p$, and $G8_p$ at 0.5 mM $MgCl_2$ were different from those at 50 mM NaCl (Fig. 7). In addition, the pAID side signals of $S1_p$, $Q6_p$, and $K9_p$ at 50 mM NaCl were not observed at 0.5 mM $MgCl_2$ (Fig. 7 and Supplementary Figs. 5a, 11d). Furthermore, regarding the H2B BS2 region, the pAID side signals of $A17_p$,

$V18_p$, $T19_p$, $A21_p$, $Q22_p$, $K23_p$, and $K24_p$ at 50 mM NaCl were not observed at 0.5 mM $MgCl_2$ (Supplementary Figs. 5a and 11d). The 112-bp DNA/pAID nucleosome was aggregated at a much lower $MgCl_2$ concentration (1 mM) as compared with the 145-bp nucleosome (4 mM). The interaction of the H2A N-tail and the H2B BS2 region with pAID may lead to inter-nucleosomal interactions via magnesium ions; in other words, the H2A and H2B N-tails of one nucleosome interact with pAID in another nucleosome bridged by magnesium ions.

## Discussion

The present NMR analysis has verified that, in the 112-bp DNA/ pAID nucleosome (Fig. 1b), the H2A N-tail and the BS2 regions of the H2B N-tail adopt asymmetric conformations, reflecting the DNA and pAID side environments (Figs. 2, 3, 4b, c, and 5a, b and Supplementary Tables 1 and 2). In contrast, the H2A C-tail and the NS, BS1, and LS regions of the H2B N-tail showed no obvious difference in the interaction and/or conformation between the two sides (Fig. 2 and Supplementary Tables 1 and 2), despite their known asymmetric environments in the 112-bp DNA/pAID

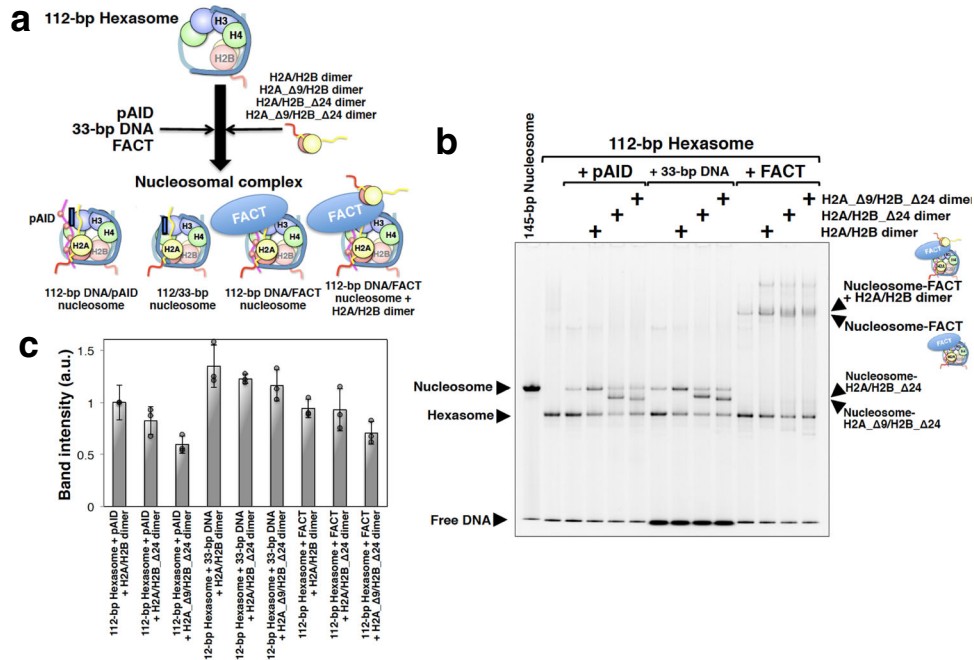

**Fig. 6 H2A and H2B N-tails play an important role in nucleosome assembly with hFACT. a** Cartoon model of the nucleosomal complex. Histone proteins and DNA are colored as follows: H2B (red), H2A (yellow), H3 (light blue), H4 (green), and DNA (light sea green). The H2A and H2B N-tails are indicated by yellow and red strings, respectively. pAID is colored magenta. Red circles labeled with P indicate phosphorylation. hFACT is colored blue. **b** Representative SYBR Gold-stained EMSAs of nucleosomal complexes in each mixture. Experiments were repeated at least three times. The full gel image is shown in Supplementary Fig. 13. In lanes 3, 5, 6, 7, 9, and 10, a faint band consistent with a full nucleosome band is observed. The reconstitution of the 112-bp hexasome by the salt dialysis method could not prevent yielding a small quantity of the 112-bp nucleosome, comprising a histone octamer and 112-bp DNA. The faint band in lanes 3, 5, and 6 is likely to correspond to its complex with additional pAID, and the faint band in lanes 7, 9, and 10 may correspond to its complex with additional 33-bp DNA. Also, the faint band in lane 11 is likely to correspond to the 112-bp nucleosome interacting with hFACT. **c** Band intensity of each nucleosomal complex in **b**. Band intensity was quantified by using Image Lab software (Bio-Rad), and the intensity of the 112-bp hexasome with pAID and H2A/H2B dimer was set to one. Averages from at least three independent experiments are shown; error bars represent SD.

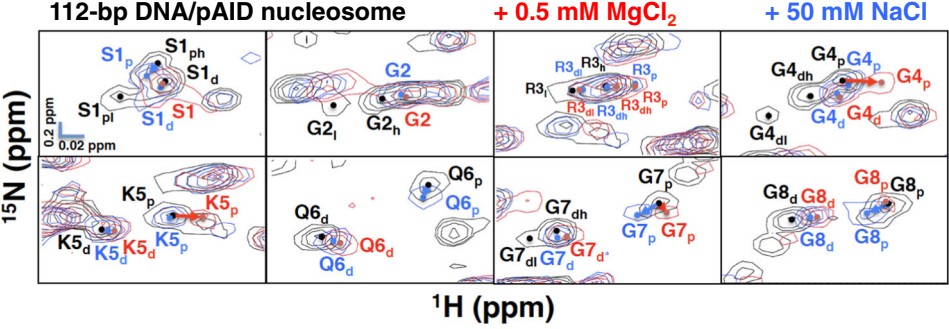

**Fig. 7 Conformational comparison of the H2A N-tail upon metal ion titration.** Expanded signal comparison of the H2A N-tail in the 112-bp DNA/pAID nucleosome at 0 M (black), 0.5 mM $MgCl_2$ (red), and 50 mM NaCl (blue). NMR signals are divided into panels by each amino acid residue, Ser1–Gly8 of H2A. Signal assignments in the 112-bp DNA/pAID nucleosome at 0 M, 0.5 mM $MgCl_2$, and 50 mM NaCl are labeled in black, red, and blue, respectively. Filled circles indicate each signal center. pAID and DNA side signals are designated by subscript p and d, respectively. High- and low-field components of the signals are designated by subscript h and l, respectively. Arrows indicate chemical shift changes of pAID side signals between 0 mM and 0.5 mM $MgCl_2$ (red) or 50 mM NaCl (blue).

nucleosome (Fig. 1b, c). These results indicate that the conformational differences reflecting the asymmetric environments of the tails arise from particular regions of H2A and H2B that show multiple interactions with DNA in the canonical nucleosome[33]. Therefore, the complicated behavior of the H2A and H2B N-tails is markedly different from the simple behavior of the H3 N-tails, where most residues show two distinct conformations reflecting the pAID and DNA sides in the 112-bp DNA/pAID nucleosome[24].

Under the physiological salt condition, the NS, BS1, and LS regions of the H2B N-tail (Glu2–Lys15 and Asp25–Lys27) dynamically interact with DNA and pAID in a similar manner (Supplementary Table 2), whereas the H2A N-tail and the BS2 region in the H2B N-tail interact more robustly with pAID than with DNA (Figs. 4d and 5c). We confirmed that these robust interactions are important for nucleosome assembly with hFACT (Fig. 6b, c). The functional conformation of the pAID side H2A and H2B N-tails (Fig. 8a) is explicitly different from that of the

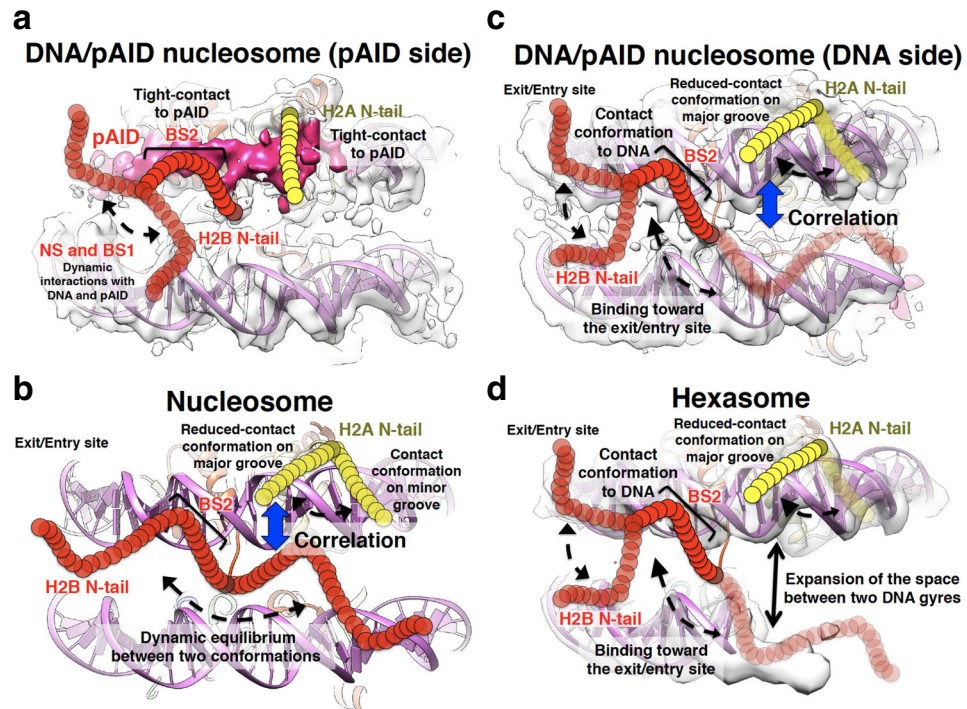

**Fig. 8 Summary of the dynamic conformations of the H2A and H2B N-tails. a–d** Dynamic structures of the H2A and H2B N-tails in the DNA/pAID nucleosome on the pAID (**a**) and DNA (**c**) sides, in nucleosome (**b**), and in hexasome (**d**), colored as in Fig. 1. In **b**, the canonical nucleosome structure (PDB ID: 2CV5) is shown. In **a**, **c**, the EM density map of the 112-bp DNA/pAID nucleosome (EMD-9639) is superimposed on the nucleosome structure (PDB ID: 2CV5) lacking the 33-bp DNA. In **d**, the EM density map of the 112-bp hexasome (EMD-6939) is superimposed on the nucleosome structure (PDB ID: 2CV5) lacking the 43-bp DNA and one H2A/H2B dimer. Yellow and red circular chains denote the H2A and H2B N-tails, respectively. Dotted arrows represent the dynamic behavior of the H2A and H2B N-tails. Black arrow represents expansion of the space between two DNA gyres in hexasome, as shown in the cryo-EM structure of the 112-bp hexasome[10]. Blue arrows represent conformational correlation between the H2A N-tail and the H2B N-tail.

pAID side H3 N-tails, where most residues adopt the relatively exposed conformation[24]. In other words, while the H3 N-tail adopts an accessible conformation in the unwrapped nucleosome with pAID, the H2A and H2B N-tails retain the nucleosomal structure by interacting with pAID. This indicates that the robust interactions of the H2A and H2B N-tails with pAID compensate for the reduced interaction between the H3 N-tail and pAID, thereby maintaining a stable nucleosome core structure with hFACT as a whole. In a recent report, mono-ubiquitination at Lys119 in the H2A C-tail was shown to impede hFACT binding on nucleosome even together with the DNA stretching[58]. Taken together, direct and indirect interactions between hFACT and each histone tail are important for hFACT binding on nucleosome, although the histone tails are disordered in the structures of nucleosome-FACT complexes[10,53–55].

Whereas the DNA side H2A N-tail in the 112-bp DNA/pAID nucleosome and 112-bp hexasome adopts mainly one reduced-contact conformation, Gly2–Gly7 of the H2A N-tails in the canonical nucleosome adopt two conformations (Fig. 4b, c and Supplementary Table 1). Previous NMR analyses and MD simulations of the canonical nucleosome have indicated that the identified contact and reduced-contact conformations of the H2A N-tail are located stably on the minor and major grooves of DNA, respectively (Fig. 8b)[33]. Therefore, it is likely that the DNA side H2A N-tail within the 112-bp DNA/pAID nucleosome and 112-bp hexasome mainly adopts a major groove location corresponding to the reduced-contact conformation (Fig. 8c, d). On the other hand, in the 112/33-bp nucleosome, the proximal H2A N-tail around the double-strand break seems to be altered from the location on the minor groove corresponding to the contact conformation (Supplementary Fig. 12a). In fact, the signals of the

contact conformation in the 145-bp nucleosome at 0 mM NaCl ($G2_l$, $G4_l$, $K5_h$, $Q6_l$, and $G7_l$) were significantly affected by introducing the double-strand break in the 112/33-bp nucleosome (Supplementary Fig. 12b). In contrast, the other signals of the reduced-contact conformation in the 145-bp nucleosome ($G2_h$, $G4_h$, $K5_l$, $Q6_h$, and $G7_h$) were invariable. This indicates that the reduced-contact conformation is stably adopted regardless of the environmental difference.

The EM structure of nucleosome complexed with the virus intasome revealed that the integrase subunit within the intasome interacts with the exposed H2A N-tail in nucleosome with a loose DNA positioning[59]. Considering our finding that the DNA side H2A N-tail mainly adopts the reduced-contact conformation in the 112-bp DNA/pAID nucleosome (Fig. 8c), we envisage that the unwrapped nucleosome with hFACT renders the DNA side H2A N-tail available for association with the intasome. Therefore, it is likely that hFACT modulates chromatin signaling and accessibility at the DNA side H2A N-tail, similarly to the pAID side H3 N-tail[24]. In line with this, hFACT has been shown to regulate HIV-1 integration by modulating the interaction between the intasome and nucleosome[60].

Regarding the DNA side H2B N-tail, the BS2 region in the 112-bp DNA/pAID nucleosome adopts the DNA contact conformation, similar to that in the 112-bp hexasome (Fig. 5a, b and Supplementary Table 2). The previous EM structure of the 112-bp hexasome (Supplementary Fig. 3a) showed expansion of the space between two DNA gyres[10], implying that the H2B N-tail is unlikely to extend in the direction opposite to the entry/exit site (Fig. 8d). Thus, the H2B N-tail in hexasome seems to adopt one conformation bound to two DNA gyres in the direction toward the entry/exit site (Fig. 8d), which corresponds to a contact state

to DNA in the BS2 region. In addition, the chemical shifts of the BS2 region in the 112-bp hexasome are roughly similar to those on the DNA side in the 112-bp DNA/pAID nucleosome (Fig. 5b). This suggests that the BS2 region on the DNA side in the 112-bp DNA/pAID nucleosome likely adopts the conformation toward the entry/exit site (Fig. 8c). This model is well consistent with the previous MD simulation that the H2A N-tail location on the major grove of DNA correlates with the H2B N-tail location toward the entry/exit site in the canonical nucleosome[33] (Fig. 8b). In the canonical nucleosome, by contrast, the BS2 region in the H2B N-tail may interact with DNA in a state of dynamic equilibrium between two conformations (Fig. 8b). Two such conformations have been previously suggested by MD simulation, where the H2B N-tail can be located equally in two directions toward the entry/exit site and toward the opposite side, but the conformations cannot be assigned to the DNA contact and reduced-contact signals observed by NMR, owing to their similar structural dynamics[33]. MD simulation also suggests that the reduced-contact conformation of the H2A N-tail on the major grove of DNA correlates with the H2B N-tail location toward the entry/exit site, but not toward the opposite side (Fig. 8b)[33]. Here we have revealed a relationship between the reduced-contact conformation of the H2A N-tail and the contact conformation of the H2B N-tail in the 112-bp DNA/pAID nucleosome and 112-hexasome (Fig. 8c, d); therefore, it may safely be said that the location of the H2B N-tail toward the entry/exit site corresponds to the contact conformation and its opposite side location corresponds to the reduced-contact conformation in the canonical nucleosome (Fig. 8b).

Previous MD simulations revealed that histone tails can adopt many conformations[25,30,61]. Such MD simulations have been often performed below the microsecond timescale, which produces an averaged NMR signal for each amino acid atom. However, dynamic conformations on the microsecond to millisecond timescale are detected as independent signals in NMR[62]. Therefore, the separate signals observed in NMR seem to reflect explicitly different binding modes. For example, the conformation of the unwrapped side H3 N-tail in the 147-bp hexasome shows many conformational ensembles in MD simulations, but was detected as one averaged signal by NMR[25].

In summary, our synergistic approach of combining the cryo-EM structure with NMR analyses has clarified obvious differences in the multiple conformations of each histone tail of H2A, H2B, and H3[24] within the unwrapped nucleosome. In addition, hFACT both proximally and distally regulates the conformations of the H2A and H2B N-tails, in contrast to regulating only the proximal H3 N-tail conformation, in the asymmetrically unwrapped nucleosome. In the future, it will be essential to incorporate these findings in the unwrapped nucleosome to our fundamental knowledge of the histone tail network and its correlations with DNA within nucleosome[23,30,33].

## Materials and methods

**Protein expression and purification**. Recombinant human histone H3 and H4 proteins were produced in *Escherichia coli* and purified using a gel filtration column and a cation exchange column[63]. Full-length histone H2A and H2B proteins were produced in *Escherichia coli* and purified using a HisTrap column (GE Healthcare) and a cation exchange column[33]. $^2$H/$^{13}$C/$^{15}$N-labeled full-length histone H2A and H2B proteins were each expressed in 100% deuterated M9 minimal medium containing $^{15}$N-ammonium chloride with $^{13}$C-glucose. Truncated mutants lacking residues Ser1–Lys9 of H2A (H2A_Δ9) and Pro1–Lys24 of H2B (H2B_Δ24) were also produced in *Escherichia coli* and purified using a gel filtration column and a cation exchange column[63]. His-tagged H2B peptide, comprising N-tail residues Pro1–Tyr37, was expressed in M9 minimal medium containing $^{15}$N-ammonium chloride with $^{13}$C-glucose. The $^{13}$C/$^{15}$N-labeled H2B peptide was purified using a HisTrap column (GE Healthcare), a gel filtration column, and a cation exchange column[33]. Histone octamer, H3/H4 tetramer, full-length H2A/H2B dimer, H2A_Δ9/H2B dimer,

H2A/H2B_Δ24 dimer, and H2A_Δ9/H2B_Δ24 dimer were reconstituted and purified using a gel filtration column[9]. Baculovirus-driven expression of the hFACT complex (co-expression of N-terminal His-tagged SPT16 and non-tagged SSRP1) in Sf9 insect cells was carried out at 27 °C for 48 h[9]. pAID protein, comprising human SPT16 residues 926–964, was expressed in *E. coli* strain BL21 (DE3) by using plasmid pColdI (pAID) with pRSFduet (Casein Kinase 2)[10]. hFACT and pAID proteins were purified using a HisTrap column (GE Healthcare), a gel filtration column, and a anion exchange column[9,10]. All DNA segments are based on the 601 nucleosome positioning sequence[64]. DNA fragments of 33-bp, 112-bp, and 145-bp were constructed and purified using a anion exchange column[9]. The 112-bp DNA/pAID nucleosome and 145-bp nucleosome were reconstituted from histone octamer, DNA fragments, and pAID by the salt dialysis method, and then purified using an anion exchange column[10]. The 112-bp hexasome was reconstituted with 112-bp DNA, H3/H4 tetramer, and an equivalent amount of H2A/H2B dimer by the salt dialysis method and purified using an anion exchange column[10].

**NMR spectroscopy**. All NMR spectra were recorded at 293 K on Bruker Avance III HD 950-MHz and 600-MHz spectrometers, equipped with cryogenically triple-resonance pulsed-field gradient cryoprobes. All spectra were processed with the program NMRPipe[65] and analyzed by the program Olivia (M. Yokochi, S. Sekiguchi & F. Inagaki, Hokkaido University, Sapporo, Japan). For backbone assignment, the 112-bp/pAID nucleosome (100 μM) containing $^2$H/$^{13}$C/$^{15}$N-labeled histone H2A and H2B in NMR buffer (20 mM HEPES-NaOH pH 7.0, 10% D$_2$O) was used. For other NMR experiments, 112-bp/pAID nucleosome (30 μM), 112-bp hexasome (70 μM), and 145-bp nucleosome (40 μM) containing $^2$H/$^{13}$C/$^{15}$N-labeled histone H2A and H2B in NMR buffer were used. Backbone resonances were assigned via the following experiments: TROSY-HNCACB, TROSY-HN(CO)CACB, TROSY-HNCO, TROSY-HN(CA)CO and 2D TROSY-$^1$H-$^{15}$N HSQC. Chemical shift differences (Δδ) were calculated according to $\Delta\delta = \{(\Delta\delta^1H)^2 + (\Delta\delta^{15}N/5)^2\}^{1/2}$.

"ΔCα-ΔCβ" values were calculated by the following equation:

$$(\Delta C\alpha - \Delta C\beta)_i = \frac{1}{3}\sum_{i=i-1}^{3}[(\delta C\alpha_{obsi} - \delta C\alpha_{refi}) - (\delta C\beta_{obsi} - \delta C\beta_{refi})]$$

where Cα$_{obs}$ and Cβ$_{obs}$ indicate measured chemical shifts and Cα$_{ref}$ and Cβ$_{ref}$ indicate random coil chemical shifts[66].

**Titration experiments**. Titration of 33-bp DNA into the 112-bp DNA/pAID nucleosome (30 μM) incorporating labeled H2A and H2B at molar ratios of 0.5:1, 1:1, and 2:1 was performed in NMR buffer at 293 K. Addition of an equivalent amount of 33-bp DNA or pAID into 112-bp hexasome incorporating labeled H2A and H2B (70 μM), H2A/H2B dimer containing $^2$H/$^{13}$C/$^{15}$N-labeled histone H2A or H2B (each 100 μM), and a $^{13}$C/$^{15}$N-labeled H2B peptide (100 μM) was performed in NMR buffer. The changes in signals caused by substrate titration were monitored by TROSY-$^1$H-$^{15}$N HSQC. Titration of NaCl (50 mM, 100 mM, 200 mM, and 300 mM) or MgCl$_2$ (0.5 mM, 1 mM, 2 mM, and 4 mM) into the 112-bp DNA/pAID nucleosome (30 μM), 112-bp hexasome (70 μM), and 145-bp nucleosome (40 μM) incorporating labeled H2A and H2B was performed by the same method.

**Electrophoretic mobility shift assays (EMSAs)**. The 112-bp hexasome (1.5 pmol) was mixed with pAID, the 33-bp DNA, or hFACT proteins (each 1.5 pmol) and either full-length H2A/H2B dimer or truncated mutant dimer (each 1.5 pmol) in a reaction buffer containing 100 mM NaCl, 20 mM Tris-HCl, pH 7.6. To clarify the nucleosomal complexes formed with hFACT, the 112-bp hexasome (1.5 pmol) was mixed with hFACT (1.5, 3.0, 6.0 pmol) and full-length H2A/H2B dimer (1.5, 3.0, 4.5 pmol) in a reaction buffer. The samples were incubated for 15 min at 30 °C, separated by electrophoresis at 4 °C on a 7.5% native-PAGE in 1× Tris-glycine buffer, and visualized by SYBR Gold nucleic acid gel stain.

**Statistics and reproducibility**. EMSA experiments were repeated three times.

**Reporting summary**. Further information on research design is available in the Nature Research Reporting Summary linked to this article.

## Data availability
The NMR data have been deposited in the Biological Magnetic Resonance Data Bank (BMRB) under accession numbers 51178 and 51177 for the 112-bp DNA/pAID nucleosome and 112-bp hexasome, respectively. The full gel images in Fig. 6b and Supplementary Fig. 10a, b are shown in Supplementary Fig. 13. The source data in Figs. 4d, 5c, and 6c are shown in Supplementary Data 1. All other data are available from the authors upon reasonable request.

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

## Acknowledgements

The authors thank Dr. Kosuke Morikawa for helpful discussion. This work was supported, in part, by a Grants-in-Aid for Scientific Research (C) (JSPS KAKENHI grant nos. JP18K06064 and JP21K06021 to Y.T.); by a Grants-in-Aid for Scientific Research on an NMR platform (grant no. 07022019 to Y.N.) from the Ministry of Education, Culture, Sports, Science and Technology (MEXT), Japan; and by a Platform Project for Supporting Drug Discovery and Life Science Research (Basis for Supporting Innovative Drug Discovery and Life Science Research; BINDS) from the Japan Agency for Medical Research and Development (AMED; grant no. JP21am0101073 to Y.N.).

## Author contributions

Conceptualization, Y.T. and Y.N.; Sample Preparation, Y.T.; Investigation, Y.T. and H.O.; Writing—Original Draft, Y.T.; Writing—Review & Editing, H.O. and Y.N.; Funding Acquisition, Y.T. and Y.N.; Resources, Y.T. and Y.N.; Supervision, Y.N.

## Competing interests

The authors declare no competing interests.
