## [Peer Review File · Communications Biology]

Reviewers' comments:

Reviewer #1 (Remarks to the Author):

In this article, Tsunaka and colleagues reported a solution-state NMR study of the conformations of H2A and H2B flexible tails interacting with pAID and DNA in the pAID-nucleosome complexes. The dynamic conformations of those tails showed that the H2B BS2 and H2A N-tail regions strongly interact with pAID on the pAID-proximal side in the partially unwrapped nucleosome complex. On the DNA side, the interactions of H2B and H2A with DNA were altered in comparison with the canonical nucleosomes. In addition, the EMSAs experiment highlighted that the H2A and H2B tails interacting with pAID was important for nucleosome assembly from hexasome.

Overall, this study includes extensive experiments, and the data are comprehensive and supports the conclusions. I recommend publishing this study in *Communications Biology* with revision that addresses the following points.

1. Based on the text, in Fig. 2 and Supplemental Fig. 1, the 112/33-bp nucleosome sample was the product of 112-bp DNA/pAID nucleosome titrated with 33 bp DNA (two-fold?), however, this information was missing in both the figure captions and figures, which will lead much confusion to the audience. This information should be stated clearly in the figures or figure captions.
2. The experimental details of how those titration were performed were missing in the manuscript. The details including concentrations, steps of titration points, ect, should be provided.
3. When titrating 33 bp DNA into the nucleosomes, what is the saturation point, 2-fold?
4. In Fig. 2, only spectra of two points (initial and last) of the titration were shown, there should be spectra of more points included (probably in the supplemental material) to show that assignments for the "112/33 bp nucleosomes" are not biased, as that the overlap and noises affect the certainty of the assignments in the current presentation in Fig. 2. For example, a second unassigned peak (blue) was observed near G8, does this peak belong to G8?
5. For many peaks shown in the figures, there are overlaps and noises that likely cause problems for identifying the centers. The authors should comment how they identified the centers for difficult regions and what was the uncertainty.
6. Line 375, "NDA" misspelling.
7. The Materials and Methods section didn't include all necessary information, complete details should be included.
8. In a few figure captions, the denotation of Arrows were note given.
9. Line 54, other references beside ref 31 and 32 should be cited here too, for example, the representative solid-state NMR studies of histone tails in nucleosomes (e.g. <https://doi.org/10.1038/s42003-020-01369-3>; <https://doi.org/10.1021/acs.jpcllett.1c01187>;) and solution-state NMR work from L. Kay group.
10. In the Discussion section, the authors mentioned "dynamic exchange between two conformations", the "dynamic equilibrium between two confirmations", etc, which suggests the dynamics property of the N-tails. In many studies like MD simulation (e.g. <https://doi.org/10.1038/s41467-021-22636-9>), there is often suggestion that the N-tails have many conformation assemblies (heterogeneous) and at the same time highly flexible (that is visible in NMR), which seems controversy to that only a few conformations were detected by NMR. Could the author comment on the timescales of the dynamic exchanges, and whether NMR contradicts with those MD results?
11. Fig. 2 caption, a space should be added between "H2Aand"
12. The literature citation format is not consistent.

Reviewer #2 (Remarks to the Author):

The article by Tsunaka et al. presents NMR analysis of histones H2A and H2B within a hexasome bound by the SPT16 acidic domain (pAID) in FACT. Based on previous work from the same authors, this article delves into the conformation of the H2A and H2B tails in this complex. FACT is a critical factor in chromatin biology, studying its interaction with histones at the structural and dynamic level is important for many aspects of biology. However, this article fails to explain how their data fits into the large body of literature available on FACT and nucleosome interactions.

The manuscript is written and driven solely by the previous work from the authors. Many (if not all) relevant recent FACT papers are not cited (work from Formosa, Li, Luger, and many more labs DOIs: 10.1038/s41586-019-1820-0, 10.1016/j.molcel.2018.06.020, etc) or on the NMR studies of histones in nucleosomes (e.g. DOI: 10.1021/jacs.8b00931). Essentially, this makes the article flawed.

For these reasons, the article should not be published as is. A complete revision of the interpretation of the NMR measurements should be considered and a broad explanation of the implications should be included.

Reviewer #3 (Remarks to the Author):

In this manuscript, Tsunaka et al. use NMR spectroscopy to characterize the structure and interactions of the H2A and H2B histone tails in a nucleosome interacting with the pAID domain of the FACT chaperone complex. In previous work, the authors have shown that 1) the two copies of the N-terminal tail of histone H3 adopt different conformations in this type of nucleosome, and 2) the H2A and H2B N-terminal tails adopt two different conformations in a wild-type nucleosome. This manuscript represents a continuation of this work. The main conclusions of this study are 1) one set of H2A and H2B N-terminal tails in the nucleosome interact strongly with pAID, 2) the second set of H2A and H2B, on the DNA side, experience altered conformations in the pAID nucleosome compared to the wild-type, and 3) the two copies of the H2A C-terminal tail adopt similar conformations.

Elucidating the interactions and dynamics of the histone tails in different nucleosome complexes is important as binding interactions might alter the accessibility of the tail and affect downstream post-translational modifications and biological events. However, as currently written, the manuscript is not ready for publication.

The differences in conformation are most pronounced in 0 mM salt conditions and the conformations of the tails seem to become much more similar upon titration of monovalent salt. Therefore, one is left to wonder whether the detected conformations are biologically significant under physiological conditions. If so, what would their possible biological role be? And what happens in the presence of relevant divalent cations such as Mg²⁺?

Upon titration with salt, the chemical shifts of the tails change. The authors interpret larger chemical shift change with tighter binding with pAID? Chemical shifts changes reflect a variety of factors, including the influence of the added salt, so what is the basis for the authors' interpretation?

In the competition experiment where a 33 bp DNA is used to displace pAID from the nucleosome, additional peaks are observed for some residues. Could this be due to interactions with the free 33 bp DNA or the displaced pAID?

Question for Fig. 6, part b, lanes with FACT. On this type of gel, is it possible to distinguish between a hexasome-FACT complex and an assembled full nucleosome-FACT complex? It appears that all bands labeled as nucleosome-FACT complex run at the same place, but one of them is clearly a hexasome-FACT complex as no H2A-H2B dimer has been added in that lane (fourth from the right). Also, for this assembly experiment, does the deletion of the H2A tail on its own impede assembly with FACT or the deletion of both H2A and H2B tails is required?

And finally, I found the paper hard to read and follow. It would have been helpful to summarize the observed changes in different spectra in a table format so that one can easily refer back to the relevant residues rather than search the text. Also, it might be helpful to add a section in the discussion or results, where each nucleosome type is compared with each other, e.g. 112 bp/pAID to wild-type, hexasome to wild-type, 33bp/112pb nucleosome to wild-type, hexasome to 112 bp/pAID etc., so that the reader can connect better with Fig. 7. I also had a really hard time distinguishing the colors in some figures, e.g. the red and yellow (?) in Figure 4.

We are very grateful to you for giving us the opportunity to revise our manuscript for publication. We believe that our revisions, along with some additional experiments (Figs. 4b–d, 5, and 7, Supplementary Figs. 2, 6, 7, 10, and 11, and Supplementary Tables 1 and 2), have sufficiently addressed all of the comments raised by the three reviewers. In responding to the comments, we have replaced some of the original figures with new figures in the revised manuscript, as detailed below. We have also deleted some figures (original Figs. 3a and 4c and Supplementary Figs. 3d and 5b), and inserted new figures and tables (Figs. 4b–d, 5, and 7, Supplementary Figs. 1, 2b, c, 3d, 6, 7, 10, and 11, and Supplementary Tables 1 and 2). Also in response to the reviewers' comments, 14 references (Refs. 31, 34–39, 52–55, 58, 61, and 62) have been added, and the descriptions in the revised manuscript have been thoughtfully improved. All of our changes are marked in red in the revised manuscript. In addition, the English in the revised manuscript has been checked and improved through an English correction service. In this response letter, the reviewers' comments are indicated in standard black, and our responses are written in blue.

Figure changes

Original Fig. 2a > New Fig. 2a

Original Fig. 3b > New Fig. 3a

Original Fig. 4a > New Supplementary Fig. 8a

Original Fig. 4b > New Fig. 4a

Original Fig. 4d > New Fig. 3b

Original Fig. 4e > New Fig. 3c

Original Fig. 5 > New Supplementary Fig. 9

Original Fig. 6a, b > New Fig. 6a, b

Original Fig. 7 > New Fig. 8

Original Supplementary Fig. 1a > New Supplementary Fig. 2a

Original Supplementary Fig. 1b > New Supplementary Fig. 2d

Original Supplementary Fig. 1c > New Supplementary Fig. 3a

Original Supplementary Fig. 1d > New Supplementary Fig. 3b

Original Supplementary Fig. 2 > New Supplementary Fig. 5

Original Supplementary Fig. 3a > New Supplementary Fig. 8b
Original Supplementary Fig. 3b > New Supplementary Fig. 8c
Original Supplementary Fig. 3c > New Supplementary Fig. 8d
Original Supplementary Fig. 5a > New Supplementary Fig. 3c
Original Supplementary Fig. 5c > New Supplementary Fig. 12b
Original Supplementary Fig. 5d > New Supplementary Fig. 12a
Original Supplementary Fig. 6 > New Supplementary Fig. 13

Reviewers' comments:

Reviewer #1 (Remarks to the Author):

In this article, Tsunaka and colleagues reported a solution-state NMR study of the conformations of H2A and H2B flexible tails interacting with pAID and DNA in the pAID-nucleosome complexes. The dynamic conformations of those tails showed that the H2B BS2 and H2A N-tail regions strongly interact with pAID on the pAID-proximal side in the partially unwrapped nucleosome complex. On the DNA side, the interactions of H2B and H2A with DNA were altered in comparison with the canonical nucleosomes. In addition, the EMSAs experiment highlighted that the H2A and H2B tails interacting with pAID was important for nucleosome assembly from hexasome.

Overall, this study includes extensive experiments, and the data are comprehensive and supports the conclusions. I recommend publishing this study in *Communications Biology* with revision that addresses the following points.

We would like to thank the reviewer for their understanding of the significance of our paper and for providing constructive comments. We have taken into account each of their comments, as described in detail in the point-by-point responses listed below.

1. Based on the text, in Fig. 2 and Supplemental Fig. 1, the 112/33-bp nucleosome sample was the product of 112-bp DNA/pAID nucleosome titrated with 33 bp DNA (two-fold?), however, this information was missing in both the figure captions and figures, which will lead much confusion to the

audience. This information should be stated clearly in the figures or figure captions.

We apologize for our insufficient description. According to this comment, we have corrected figures (Fig. 2a and Supplementary Figs. 2, 8d, and 12b) and added the relevant information to the figure captions (Fig. 2 and Supplementary Figs. 2, 8d, and 12b) of the revised manuscript as follows.

Caption of Fig.2

“Fig. 2. Comparison of extended HSQC spectra of the H2A and H2B tails between the 112-bp DNA/pAID nucleosome alone (red) and with the two-fold addition of 33-bp DNA (blue).

(a, b) NMR signals are divided into panels by amino acid residue of H2A (a) and H2B (b). Signal assignments in the 112-bp DNA/pAID nucleosome alone and with the two-fold addition of DNA at 0 mM NaCl are labeled in red and blue, respectively. Two-fold addition of 33-bp DNA to the 112-bp DNA/pAID nucleosome leads to a double-strand break nucleosome wrapped by 112-bp and 33-bp DNA (112/33-bp nucleosome), as shown in Supplementary Fig. 2a.”

Caption of Supplementary Fig. 2

“Supplementary Figure 2. 2D ^1H - ^{15}N HSQC spectra of the H2A and H2B tails upon 33-bp DNA titration.

(a) Cartoon model of the 112/33-bp nucleosome, formed by the addition of 33-bp DNA up to an excess over the 112-bp DNA/pAID nucleosome. Cryo-EM structure of the 112-bp DNA/pAID nucleosome is shown as in Fig. 1a. Histone proteins and DNA are colored as follows: H2A (yellow), H2B (red), H3 (light blue), H4 (green), and DNA (light sea green). The H2A and H2B N-tails are indicated by yellow and red strings, respectively. pAID is colored magenta. Red circles labeled with P indicate phosphorylation.

(b) Spectral superposition of the 2D ^1H - ^{15}N HSQC spectra of the 112-bp DNA/pAID nucleosome without (black) and with the addition of 33-bp DNA at a molar ratio of 1:0.5 (red). Signal assignments in the 112-bp DNA/pAID nucleosome upon titration with 33-bp DNA at a molar ratio of 1:0.5 are

labeled in red. pAID and DNA side signals are designated by subscript p and d, respectively. Blue and orange lines indicate residues of H2A and H2B, respectively.

(c) Spectral superposition of the 2D ^1H - ^{15}N HSQC spectra of the 112-bp DNA/pAID nucleosome upon titration with 33-bp DNA at molar ratios of 1:0.5 (red) and 1:1 (cyan). Signal assignments in the 112-bp DNA/pAID nucleosome upon an equivalent addition of 33-bp DNA are labeled in cyan.

(d) Spectral superposition of the 2D ^1H - ^{15}N HSQC spectra of the 112-bp DNA/pAID nucleosome upon titration with 33-bp DNA at molar ratios of 1:1 (cyan) and 1:2 (orange). Signal assignments in the 112-bp DNA/pAID nucleosome upon two-fold addition of 33-bp DNA are labeled in orange.”

Caption of Supplementary Fig. 8d

“(d) Expanded signal comparison of Ala126–Gly128 in H2A C-tails between the 112-bp hexasome upon the addition of an equivalent amount of pAID (black) and the 112-bp DNA/pAID nucleosome alone (red) (upper panel) or with the two-fold addition of 33-bp DNA (112/33-bp nucleosome, blue) (lower panel).”

Caption of Supplementary Fig. 12b

“(b) Expanded signal comparison of Gly2–Gly7 in the H2A N-tail between the 145-bp nucleosome (black) and the two-fold addition of 33-bp DNA to the 112-bp DNA/pAID nucleosome (112/33-bp nucleosome, blue) at 0 mM NaCl.”

2.The experimental details of how those titration were performed were missing in the manuscript. The details including concentrations, steps of titration points, etc, should be provided.

We apologize for our insufficient description. To clarify more precisely this point, we have added detailed methods for the titration experiments to the revised manuscript as follows.

Line 656

“Titration experiments

Titration of 33-bp DNA into the 112-bp DNA/pAID nucleosome (30 μ M) incorporating labeled H2A and H2B at molar ratios of 0.5:1, 1:1, and 2:1 was performed in NMR buffer at 293 K. Addition of an equivalent amount of 33-bp DNA or pAID into 112-bp hexasome incorporating labeled H2A and H2B (70 μ M), H2A/H2B dimer containing $^2\text{H}/^{13}\text{C}/^{15}\text{N}$ -labeled histone H2A or H2B (each 100 μ M), and a $^{13}\text{C}/^{15}\text{N}$ -labeled H2B peptide (100 μ M) was performed in NMR buffer. The changes in signals caused by substrate titration were monitored by TROSY- ^1H - ^{15}N HSQC. Titration of NaCl (50 mM, 100 mM, 200 mM, and 300 mM) or MgCl_2 (0.5 mM, 1 mM, 2 mM, and 4 mM) into the 112-bp DNA/pAID nucleosome (30 μ M), 112-bp hexasome (70 μ M), and 145-bp nucleosome (40 μ M) incorporating labeled H2A and H2B was performed by the same method.”

3. When titrating 33 bp DNA into the nucleosomes, what is the saturation point, 2-fold?

We apologize for our insufficient representation. We have added the HSQC spectra of the 112-bp DNA/pAID nucleosome upon titration with 33-bp DNA at molar ratios of 1:0.5 and 1:1 to the revised manuscript (Supplementary Fig. 2b–d). The signals of Ser1, Arg3–Gly8, and Ala126–Gly128 of H2A and Lys16–Gly26 of H2B were significantly changed upon the addition of an equivalent amount of DNA (Supplementary Fig. 2c), and then more or less remained the same up to the two-fold addition of DNA (Supplementary Fig. 2d). This result suggests that the saturation point is almost reached at the equivalent addition of DNA. To clarify this point, we have revised the following sentence, and added a new figure (Supplementary Fig. 2) to the revised manuscript.

Line 137

“To determine which are the pAID side signals of the H2A and H2B tails, we titrated 33-bp DNA into the 112-bp DNA /pAID nucleosome incorporating labeled H2A and H2B at molar ratios of 0.5:1, 1:1, and 2:1 (Supplementary

Fig. 2a–d). The signals of Ser1, Arg3–Gly8, and Ala126–Gly128 of H2A, and Lys16–Gly26 of H2B were significantly changed upon the addition of an equivalent amount of DNA (Supplementary Fig. 2c), and remained almost the same up to the two-fold addition of DNA (Supplementary Fig. 2d). This result suggests that the saturation point is more or less reached on equivalent addition, and the excess 33-bp DNA present after the two-fold addition of DNA hardly affects the chemical shifts.”

4. In Fig. 2, only spectra of two points (initial and last) of the titration were shown, there should be spectra of more points included (probably in the supplemental material) to show that assignments for the “112/33 bp nucleosomes” are not biased, as that the overlap and noises affect the certainty of the assignments in the current presentation in Fig. 2. For example, a second unassigned peak (blue) was observed near G8, does this peak belong to G8?

In response to the reviewer’s comment, we have included the HSQC spectra of the 112-bp DNA/pAID nucleosome upon titration with 33-bp DNA at molar ratios of 1:0.5 and 1:1 in Supplementary Fig. 2 of the revised manuscript.

Regarding Gly8 of H2A in the 112-bp DNA/pAID nucleosome upon titration with 33-bp DNA, we could not find the second unassigned peak in Fig. 2a (blue). As described below, we identified the signal centers from peaks in slice representations of the signals (see Supplementary Fig. 1 in the revised manuscript). According to this approach, the obvious second peak was not observed in the signals for Gly8 (Figure for reviewer).

Figure for reviewer. Slice representation of Gly8 of H2A in the 112/33-bp nucleosome.

One ^1H spectrum at the position of the horizontal line on the ^1H - ^{15}N spectrum is shown. The vertical line on the ^1H - ^{15}N spectrum indicates the signal peak of the G8 signal. The center (filled circle) was identified from the peak in slice representation of the signal.

5. For many peaks shown in the figures, there are overlaps and noises that likely cause problems for identifying the centers.

The authors should comment how they identified the centers for difficult regions and what was the uncertainty.

We apologize for our insufficient description. We identified the signal centers from peaks in slice representations of the doublet and triplet signals, as now shown in Supplementary Fig. 1 of the revised manuscript.

Supplementary Fig. 1

“Supplementary Figure 1. Slice representations of doublet and triplet signals of the 112-bp DNA/pAID nucleosome.

NMR signals are divided into panels by amino acid residue. Filled circles and numbers indicate each signal center. One, two, or three ^1H spectra at the positions of the horizontal line on each ^1H - ^{15}N spectrum are shown. Vertical lines on the ^1H - ^{15}N spectrum indicate the peaks of the doublet or triplet

signal. Signal centers are identified from peaks in the slice representations of signals.”

6.Line 375, “NDA” misspelling.

We have corrected this error.

7.The Materials and Methods section didn’t include all necessary information, complete details should be included.

According to the reviewer’s suggestion, we have added detailed methods to the Materials and Methods section of the revised manuscript as follows.

Line 641

“For backbone assignment, the 112-bp/pAID nucleosome (100 μ M) containing $^2\text{H}/^{13}\text{C}/^{15}\text{N}$ -labeled histone H2A and H2B in NMR buffer (20 mM HEPES-NaOH pH 7.0, 10% D_2O) was used. For other NMR experiments, 112-bp/pAID nucleosome (30 μ M), 112-bp hexasome (70 μ M), and 145-bp nucleosome (40 μ M) containing $^2\text{H}/^{13}\text{C}/^{15}\text{N}$ -labeled histone H2A and H2B in NMR buffer were used.”

Line 674

“To clarify the nucleosomal complexes formed with hFACT, the 112-bp hexasome (1.5 pmol) was mixed with hFACT (1.5, 3.0, 6.0 pmol) and full-length H2A/H2B dimer (1.5, 3.0, 4.5 pmol) in a reaction buffer.”

8.In a few figure captions, the denotation of Arrows were note given.

We have now defined the arrows in the captions of Figs. 2, 8, and Supplementary Fig. 12a, b, as follows.

Caption of Fig. 2

“Arrows represent chemical shift changes upon the addition of 33-bp DNA.”

Caption of Fig. 8

“Black arrow represents expansion of the space between two DNA gyres in hexasome. Blue arrows represent conformational correlation between the H2A N-tail and the H2B N-tail.”

Caption of Supplementary Fig. 12a

“Arrow represents the dynamic behavior of the H2A N-tail.”

Caption of Supplementary Fig. 12b

“To aid visualization, arrows connect signals of the contact conformation of the 145-bp nucleosome (G2_l, K5_h, Q6_l, and G7_l) to the corresponding signals of the 112/33-bp nucleosome (G2_l, K5_l, Q6_l, and G7_l).”

9.Line 54, other references beside ref 31 and 32 should be cited here too, for example, the representative solid-state NMR studies of histone tails in nucleosomes (e.g. <https://doi.org/10.1038/s42003-020-01369-3>; <https://doi.org/10.1021/acs.jpcclett.1c01187>) and solution-state NMR work from L. Kay group.

We apologize for our insufficient citation. We have added new references (Refs. 31 and 34–39) and corrected the descriptions on page 2 of the revised manuscript, as suggested by the reviewer.

Line 52

“At present, NMR analysis is the only method available to directly measure the dynamic ensembles of disordered tails within nucleosome³¹⁻³⁵. In addition, the structural dynamics of the histone core region have been characterized by methyl-based NMR spectroscopy³⁶⁻³⁹ and solid-state NMR³¹.”

New references

31. Shi, X., Prasanna, C., Soman, A., Pervushin, K. & Nordenskiöld, L. Dynamic networks observed in the nucleosome core particles couple the histone globular domains with DNA. *Commun Biol* 3, 639 (2020).
34. Zhou, B.-R. *et al.* Distinct Structures and Dynamics of Chromatosomes with Different Human Linker Histone Isoforms. *Mol Cell* 81, 166–182 (2021).
35. Zandian, M. *et al.* Conformational Dynamics of Histone H3 Tails in Chromatin. *J Phys Chem Lett* 12, 6174–6181 (2021).
36. Kato, H. *et al.* Architecture of the high mobility group nucleosomal protein 2-nucleosome complex as revealed by methyl-based NMR. *Proc Natl Acad Sci USA* 108, 12283–12288 (2011).
37. Sinha, K. K., Gross, J. D. & Narlikar, G. J. Distortion of histone octamer core promotes nucleosome mobilization by a chromatin remodeler. *Science* 355, eaaa3761 (2017).
38. Kitevski-LeBlanc, J. L. *et al.* Investigating the Dynamics of Destabilized Nucleosomes Using Methyl-TROSY NMR. *J Am Chem Soc* 140, 4774–4777 (2018).
39. Abramov, G., Velyvis, A., Rennella, E., Wong, L. E. & Kay, L. E. A methyl-TROSY approach for NMR studies of high-molecular-weight DNA with application to the nucleosome core particle. *Proc Natl Acad Sci USA* 117, 12836–12846 (2020).

10. In the Discussion section, the authors mentioned “dynamic exchange between two conformations”, the “dynamic equilibrium between two conformations”, etc, which suggests the dynamics property of the N-tails. In many studies like MD simulation (e.g. <https://doi.org/10.1038/s41467-021-22636-9>), there is often suggestion that the N-tails have many conformation assemblies (heterogeneous) and at the same time highly flexible (that is visible in NMR), which seems controversy to that only a few conformations were detected by NMR. Could the author comment on the timescales of the dynamic exchanges, and whether NMR contradicts with those MD results?

We appreciate that the histone tails can adopt many conformations in MD simulations. The MD simulations have been often performed below the microsecond time scale, which produces an averaged NMR signal for each amino acid atom. However, dynamic conformations on the microsecond to millisecond time scale are detected as independent signals in NMR (Ref. 62). Thus, we think that some distinct signals in NMR seem to reflect explicitly different binding modes, such as contact and reduced contact to DNA and pAID. For example, the reduced-contact state of the histone tail to DNA may have many conformational ensembles in MD. In fact, a recent report revealed that the reduced-contact conformation of the unwrapped side H3 N-tail in the 147-bp hexasome has many conformational ensembles in MD simulations, but was detected as one averaged signal in NMR (Ref. 25). Therefore, we consider that our results from NMR do not contradict the previous MD results. To clarify this point, we have added the following text and new references (Refs. 61 and 62) to the revised manuscript.

Line 585

“Previous MD simulations revealed that histone tails can adopt many conformations^{25,30,61}. Such MD simulations have been often performed below the microsecond timescale, which produces an averaged NMR signal for each amino acid atom. However, dynamic conformations on the microsecond to millisecond timescale are detected as independent signals in NMR⁶². Therefore, the separate signals observed in NMR seem to reflect explicitly different binding modes. For example, the conformation of the unwrapped side H3 N-tail in the 147-bp hexasome shows many conformational ensembles in MD simulations, but was detected as one averaged signal by NMR²⁵.”

New references

61. Armeev, G. A., Kniazeva, A. S., Komarova, G. A., Kirpichnikov, M. P. & Shaytan, A. K. Histone dynamics mediate DNA unwrapping and sliding in nucleosomes. *Nat. Commun.* 12, 2387 (2021).

62. Palmer, A. G., III, Kroenke, C. D. & Loria, J. P. Nuclear Magnetic Resonance Methods for Quantifying Microsecond-to-Millisecond Motions in Biological Macromolecules. *Methods Enzymol* 339, 204–238 (2001).

11. Fig. 2 caption, a space should be added between “H2Aand”

We have now added space between “H2Aand” in the caption of Fig. 2.

12. The literature citation format is not consistent.

We apologize for the inconsistency in the citation format. We have corrected the citation format in the revised manuscript.

Reviewer #2 (Remarks to the Author):

The article by Tsunaka et al. presents NMR analysis of histones H2A and H2B within a hexasome bound by the SPT16 acidic domain (pAID) in FACT. Based on previous work from the same authors, this article delves into the conformation of the H2A and H2B tails in this complex.

In our manuscript, we have mainly clarified the detailed conformations of the H2A and H2B tails within the 112-bp nucleosome bound to pAID (112-bp DNA/pAID nucleosome), and used NMR data for the 112-bp hexasome and the 145-bp nucleosome as a reference.

1. FACT is a critical factor in chromatin biology, studying its interaction with histones at the structural and dynamic level is important for many aspects of biology. However, this article fails to explain how their data fits into the large body of literature available on FACT and nucleosome interactions. The manuscript is written and driven solely by the previous work from the authors.

Many (if not all) relevant recent FACT papers are not cited (work from Formosa, Li, Luger, and many more labs DOIs: 10.1038/s41586-019-1820-0,

10.1016/j.molcel.2018.06.020, etc) or on the NMR studies of histones in nucleosomes (e.g. DOI: 10.1021/jacs.8b00931). Essentially, this makes the article flawed.

For these reasons, the article should not be published as is. A complete revision of the interpretation of the NMR measurements should be considered and a broad explanation of the implications should be included.

We apologize for our insufficient descriptions and citations, which led to the reviewer's critical comments. We have additionally referred to other relevant papers in the revised manuscript, as suggested by the reviewer (Refs. 31, 34–39, 52–55, and 58), and have corrected the text in the revised manuscript as indicated below. We believe that the revised manuscript sufficiently clarifies our new insights in the context of relevant recent studies on FACT.

Line 52

“At present, NMR analysis is the only method available to directly measure the dynamic ensembles of disordered tails within nucleosome³¹⁻³⁵. In addition, the structural dynamics of the histone core region have been characterized by methyl-based NMR spectroscopy³⁶⁻³⁹ and solid-state NMR³¹.”

Line 69

“In agreement with that finding, human SPT16 has been shown to displace H2A-H2B dimers from nucleosome to unwrap the nucleosomal DNA together with DNA stretching⁵². Second, we revealed the cryo-electron microscopy (cryo-EM) structure of a partially unwrapped nucleosome complexed with hFACT, in which 112-bp DNA and pAID are asymmetrically wrapped around the histone octamer (Fig. 1a, 112-bp DNA/pAID nucleosome)¹⁰. This structure highlights that pAID of hFACT retains the nucleosome core structure instead of DNA. Similar interactions between FACT and unwrapped nucleosome have been recently observed⁵³⁻⁵⁵; however, the histone tails are not visualized in those cryo-EM structures. For the 112-bp DNA/pAID nucleosome, our previous NMR study clarified that the H3 N-tails, which are invisible in the cryo-EM structure, adopt two distinct conformations reflecting their

asymmetric locations; a conformation of contact to DNA, as in the canonical nucleosome where the H3 N-tail is buried in two DNA gyres (DNA side); and a conformation of reduced contact to DNA and pAID (pAID side)²⁴.”

Line 516

“In other words, while the H3 N-tail adopts an accessible conformation in the unwrapped nucleosome with pAID, the H2A and H2B N-tails retain the nucleosomal structure by interacting with pAID. This indicates that the robust interactions of the H2A and H2B N-tails with pAID compensate for the reduced interaction between the H3 N-tail and pAID, thereby maintaining a stable nucleosome core structure with hFACT as a whole. In a recent report, mono-ubiquitination at Lys119 in the H2A C-tail was shown to impede hFACT binding on nucleosome even together with the DNA stretching⁵⁸. Taken together, direct and indirect interactions between hFACT and each histone tail are important for hFACT binding on nucleosome, although the histone tails are disordered in the structures of nucleosome-FACT complexes^{10,53-55}.”

New references

31. Shi, X., Prasanna, C., Soman, A., Pervushin, K. & Nordenskiöld, L. Dynamic networks observed in the nucleosome core particles couple the histone globular domains with DNA. *Commun Biol* 3, 639 (2020).
34. Zhou, B.-R. *et al.* Distinct Structures and Dynamics of Chromatosomes with Different Human Linker Histone Isoforms. *Mol Cell* 81, 166–182 (2021).
35. Zandian, M. *et al.* Conformational Dynamics of Histone H3 Tails in Chromatin. *J Phys Chem Lett* 12, 6174–6181 (2021).
36. Kato, H. *et al.* Architecture of the high mobility group nucleosomal protein 2-nucleosome complex as revealed by methyl-based NMR. *Proc Natl Acad Sci USA* 108, 12283–12288 (2011).
37. Sinha, K. K., Gross, J. D. & Narlikar, G. J. Distortion of histone octamer core promotes nucleosome mobilization by a chromatin remodeler. *Science* 355, eaaa3761 (2017).

38. Kitevski-LeBlanc, J. L. *et al.* Investigating the Dynamics of Destabilized Nucleosomes Using Methyl-TROSY NMR. *J Am Chem Soc* 140, 4774–4777 (2018).
39. Abramov, G., Velyvis, A., Rennella, E., Wong, L. E. & Kay, L. E. A methyl-TROSY approach for NMR studies of high-molecular-weight DNA with application to the nucleosome core particle. *Proc Natl Acad Sci USA* 117, 12836–12846 (2020).
52. Chen, P. *et al.* Functions of FACT in Breaking the Nucleosome and Maintaining Its Integrity at the Single-Nucleosome Level. *Mol Cell* 71, 284–293 (2018).
53. Liu, Y. *et al.* FACT caught in the act of manipulating the nucleosome. *Nature* 577, 426–431 (2020).
54. Farnung, L., Ochmann, M., Engholm, M. & Cramer, P. Structural basis of nucleosome transcription mediated by Chd1 and FACT. *Nat Struct Mol Biol* 28, 382–387 (2021).
55. Sivkina, A. L. *et al.* Electron microscopy analysis of ATP-independent nucleosome unfolding by FACT. *Commun Biol* 5, 2 (2022).
58. Wang, Y.-Z. *et al.* H2A mono-ubiquitination differentiates FACT's functions in nucleosome assembly and disassembly. *Nucleic Acids Res* 50, 833–846 (2022).

Reviewer #3 (Remarks to the Author):

In this manuscript, Tsunaka et al. use NMR spectroscopy to characterize the structure and interactions of the H2A and H2B histone tails in a nucleosome interacting with the pAID domain of the FACT chaperone complex. In previous work, the authors have shown that 1) the two copies of the N-terminal tail of histone H3 adopt different conformations in this type of nucleosome, and 2) the H2A and H2B N-terminal tails adopt two different conformations in a wild-type nucleosome. This manuscript represents a continuation of this work. The main conclusions of this study are 1) one set of H2A and H2B N-terminal tails in the nucleosome interact strongly with pAID, 2) the second set of H2A and H2B, on the DNA side, experience altered conformations in the pAID nucleosome compared to the wild-type, and 3) the

two copies of the H2A C-terminal tail adopt similar conformations.

Elucidating the interactions and dynamics of the histone tails in different nucleosome complexes is important as binding interactions might alter the accessibility of the tail and affect downstream post-translational modifications and biological events. However, as currently written, the manuscript is not ready for publication.

We greatly appreciate the reviewer's full understanding of our paper, and thank them for providing insightful and useful comments. We have incorporated their suggestions within the revised manuscript as much as possible. The details are described in the point-by-point responses listed below.

1. The differences in conformation are most pronounced in 0 mM salt conditions and the conformations of the tails seem to become much more similar upon titration of monovalent salt. Therefore, one is left to wonder whether the detected conformations are biologically significant under physiological conditions. If so, what would their possible biological role be? And what happens in the presence of relevant divalent cations such as Mg^{2+} ?

We appreciate the reviewer's significant comment. In response, we have clarified, under the physiological salt condition (100 mM NaCl), the detailed conformations of the H2A N-tail, C-tail, and the H2B N-tail within the 112-bp DNA/pAID nucleosome, in comparison to those of the 145-bp nucleosome and 112-bp hexasome in the revised manuscript (Figs. 4b–d and 5a–c, Supplementary Figs. 5b, 6b, and 7b, and Supplementary Tables 1 and 2), as follows.

H2A N-tail: Line 310

“Even under the physiological salt condition (100 mM NaCl), the pAID side signals (S1_p, R3_p, K5_p, Q6_p, G7_p, and G8_p) were observed in the 112-bp DNA/pAID nucleosome, but not in the 145-bp nucleosome or the 112-bp

hexasome (Fig. 4b, c and Supplementary Table 1). The chemical shift changes at S1_p, R3_p, K5_p, Q6_p, G7_p, and G8_p on the pAID side between 100 mM and 300 mM NaCl were significantly larger than the corresponding changes on the DNA side (S1_d, R3_{dh, dl}, K5_d, Q6_d, G7_d, and G8_d) (Fig. 4d), indicating that, under the physiological salt condition, the pAID side H2A N-tail more frequently contacts pAID as compared with the DNA side residues contacting DNA.

At 100 mM NaCl, we compared the DNA side signals for the H2A N-tail among the 112-bp DNA/pAID nucleosome, 112-bp hexasome, and 145-bp nucleosome (Fig. 4b, c and Supplementary Table 1). Doublet signals of the H2A N-tail were observed in the 145-bp nucleosome: one signal (G2_i, G4_i, K5_h, Q6_i, and G7_i) and the other (G2_h, G4_h, K5_i, Q6_h, and G7_h) reflecting the DNA contact and reduced-contact conformations of the H2A N-tail, respectively³³ (Fig. 4b and Supplementary Table 1). In contrast, the DNA side signals of the H2A N-tail in the 112-bp DNA/pAID nucleosome at 100 mM NaCl were observed as a singlet signal (G2, G4, K5_d, Q6_d, G7_d), corresponding to the reduced-contact conformation signals in the 145-bp nucleosome (G2_h, G4_h, K5_i, Q6_h, and G7_h)³³ (Fig. 4b and Supplementary Table 1). Interestingly, these signals also corresponded to the H2A N-tail signals in the 112-bp hexasome at 100 mM NaCl (S1–G8) (Fig. 4c and Supplementary Table 1). These results suggest that, under the physiological salt condition, the H2A N-tail on the DNA side of the 112-bp DNA/pAID nucleosome, as well as in the hexasome, mainly adopts the reduced-contact conformation observed in the 145-bp nucleosome.”

H2A C-tail: Line 362

“Furthermore, at 100 mM NaCl, the signals observed for the H2A C-tail in the 112-bp DNA/pAID nucleosome (E121, S122, K125, A126, K127_h, G128_h, K129_h) were equivalent to the corresponding signals in the 145-bp nucleosome and 112-bp hexasome (E121, S122, K125–K129) (Supplementary Figs. 5b, 6b, and 7b and Supplementary Table 1).”

H2B BS2 region: Line 392

“Regarding the H2B BS2 region, the pAID side signals were observed in the 112-bp DNA/pAID nucleosome even under the physiological salt condition (K16_p, A17_p, V18_p, A21_p, and K23_p), but not in the 145-bp nucleosome or the 112-bp hexasome (Fig. 5a, b and Supplementary Table 2). The chemical shift changes of the BS2 residues on the pAID side (K16_p, A17_p, V18_p, A21_p, and K23_p) between 100 and 300 mM NaCl were slightly larger than the corresponding changes on the DNA side (K16_d, A17_d, V18_d, A21_d, and K23_{dh}, dI) (Fig. 5c), indicating that the BS2 region in the H2B N-tail has increased populations of the contact states with pAID relative to the contact states with DNA under the physiological salt condition.

Next, we compared the DNA side signals for the H2B BS2 region among the 112-bp DNA/pAID nucleosome, 112-bp hexasome, and 145-bp nucleosome at 100 mM NaCl (Fig. 5a, b and Supplementary Table 2). Doublet signals were observed for BS2 in the 145-bp nucleosome (A17_{h, l}, V18_{h, l}, T19_{h, l}, A21_{h, l}, Q22_{h, l}, K23_{h, l}, and K24_{h, l}), reflecting the contact and reduced-contact conformations of the BS2 region to DNA³³ (Fig. 5a and Supplementary Table 2). In contrast, the DNA side signals of BS2 in the 112-bp DNA/pAID nucleosome at 100 mM NaCl were observed as a singlet signal (K16_d, A17_d, V18_d, T19, K20, A21_d, Q22, and K24), corresponding to the higher-field signals in the 145-bp nucleosome (K16, A17_h, V18_h, T19_h, K20, A21_h, Q22_h, and K24_h), except for doublet signals at Lys23 (Fig. 5a and Supplementary Table 2). These DNA side signals also corresponded to the signals in the 112-bp hexasome at 100 mM NaCl (K16 and V18–K24), except for Ala17 (Fig. 5b and Supplementary Table 2). In the 145-bp nucleosome, the signals of A17_h, V18_h, T19_h, and Q22_h corresponded to the DNA contact conformation, but not the reduced-contact conformation³³. Therefore, the H2B BS2 region on the DNA side in the 112-bp DNA/pAID nucleosome, as well as in the hexasome, adopts the DNA contact conformation in the 145-bp nucleosome under the physiological salt condition.”

In addition, EMSA experiments at 100 mM NaCl have revealed that the robust interactions of the both H2A and H2B N-tails with pAID are important for nucleosome assembly with FACT under the physiological salt

condition (Fig. 6 and Supplementary Fig. 10). On the pAID side in the 112-bp DNA/pAID nucleosome, whereas the H3 N-tail adopts an accessible conformation, the H2A and H2B N-tails retain their nucleosomal structure with pAID. Therefore, we consider that the robust interactions of the H2A and H2B N-tails with pAID compensate for the reduced interaction between the H3 N-tail and pAID, thereby maintaining a stable nucleosome core structure with FACT. To clarify this point, we have added the following text to the revised manuscript.

Line 516

“In other words, while the H3 N-tail adopts an accessible conformation in the unwrapped nucleosome with pAID, the H2A and H2B N-tails retain the nucleosomal structure by interacting with pAID. This indicates that the robust interactions of the H2A and H2B N-tails with pAID compensate for the reduced interaction between the H3 N-tail and pAID, thereby maintaining a stable nucleosome core structure with hFACT as a whole.”

Regarding the effect of divalent cations, we have additionally measured the chemical shift perturbations upon titration of MgCl_2 into the 112-bp DNA/pAID nucleosome and 145-bp nucleosome (Fig. 7 and Supplementary Fig. 11). To describe these data in the revised manuscript, we have added the following new section.

Line 465

“Effect of divalent cations on the 112-bp DNA/pAID nucleosome

To investigate the effect of divalent cations, we measured the chemical shift perturbations upon titration of MgCl_2 into the 112-bp DNA/pAID nucleosome and 145-bp nucleosome (Supplementary Fig. 11). Signals of the 145-bp nucleosome were observed at 0.5, 1, and 2 mM MgCl_2 , and lost at 4 mM MgCl_2 owing to severe aggregation (Supplementary Fig. 11a–c). The changes in the signals of the 145-bp nucleosome observed at 2 mM MgCl_2 were roughly equivalent to those observed at 100 mM NaCl (Supplementary Figs. 7b and 11c). Signals of the 112-bp DNA/pAID nucleosome were observed at

0.5 mM MgCl₂, and lost at 1 mM MgCl₂ owing to severe aggregation (Supplementary Fig. 11d). The changes in the signals of the 112-bp DNA/pAID nucleosome observed at 0.5 mM MgCl₂ were roughly equivalent to those observed at 50 mM NaCl (Supplementary Figs. 5a and 11d), except for some signals on the pAID side in the H2A N-tail; that is, the signal shifts of G_{4p}, K_{5p}, G_{7p}, and G_{8p} at 0.5 mM MgCl₂ were different from those at 50 mM NaCl (Fig. 7). In addition, the pAID side signals of S_{1p}, Q_{6p}, and K_{9p} at 50 mM NaCl were not observed at 0.5 mM MgCl₂ (Fig. 7 and Supplementary Figs. 5a and 11d). Furthermore, regarding the H2B BS2 region, the pAID side signals of A_{17p}, V_{18p}, T_{19p}, A_{21p}, Q_{22p}, K_{23p}, and K_{24p} at 50 mM NaCl were not observed at 0.5 mM MgCl₂ (Supplementary Figs. 5a and 11d). The 112-bp DNA/pAID nucleosome was aggregated at a much lower MgCl₂ concentration (1 mM) as compared with the 145-bp nucleosome (4 mM). The interaction of the H2A N-tail and the H2B BS2 region with pAID may lead to inter-nucleosomal interactions via magnesium ions; in other words, the H2A and H2B N-tails of one nucleosome interact with pAID in another nucleosome bridged by magnesium ions.”

2. Upon titration with salt, the chemical shifts of the tails change. The authors interpret larger chemical shift change with tighter binding with pAID? Chemical shifts changes reflect a variety of factors, including the influence of the added salt, so what is the basis for the authors' interpretation?

We appreciate the reviewer's important comment. We infer from our findings that, at high salt concentrations, the electrostatic interactions between the histone tails and DNA or pAID are weakened, and thus the populations of their contact states with DNA or pAID will be reduced. The pAID and DNA side signals of the H2A N-tail and the H2B BS2 region in the 112-bp DNA/pAID nucleosome at 0 and 50 mM NaCl were converged or approached each other at 300 mM NaCl (Figs. 3c and 4a), indicating that these N-tails are almost dissociated from both DNA and pAID. Therefore, we consider that the larger chemical shift changes of the pAID side signals

observed upon salt titration indicate that the populations of the contact states with pAID are higher than those of the contact states with DNA in the H2A N-tail and the H2B BS2 region. In addition, deletion of both N-tails of H2A and H2B impedes nucleosome assembly with pAID, but not with DNA (Fig. 6), confirming that the robust interactions of both the H2A and H2B N-tails with pAID are important for nucleosome assembly with FACT. To clarify this point, we have added the following sentences.

Line 250

“To confirm the idea that the pAID side signals are not observed at 0 mM NaCl because of the tight interaction of BS2 with pAID, we measured the chemical shift perturbations upon titration of NaCl at 50, 100, 200, and 300 mM into the 112-bp DNA/pAID nucleosome (Supplementary Fig. 5a–d) and 112-bp hexasome (Supplementary Fig. 6a–d). At high salt concentrations, electrostatic interactions between the histone tails and DNA or pAID are weakened, and thus the populations of their contact states with DNA or pAID will be reduced.”

Line 457

“This highlights that the robust interactions of both the H2A and H2B N-tails with pAID are important for nucleosome assembly from hexasome, but are not important in the interaction with DNA. This agrees well with the NMR observation that the populations of the contact states with pAID are higher than those of the contact states with DNA in the H2A N-tail and the H2B BS2 region of the 112-bp DNA/pAID nucleosome under the physiological salt condition (Figs. 4d and 5c).”

3. In the competition experiment where a 33 bp DNA is used to displace pAID from the nucleosome, additional peaks are observed for some residues. Could this be due to interactions with the free 33 bp DNA or the displaced pAID?

We appreciate that, in comparison with the 145-bp nucleosome without pAID, some additional signals were observed in the 112-bp DNA/pAID

nucleosome with the two-fold addition of 33-bp DNA (R3_i, A126_i, K127_i, and G128_i of H2A) (Supplementary Figs. 2d, 3c, 8d, and 12b). In the 112-bp hexasome, the signals of A126_i, K127_i, and G128_i were observed after the addition of an equivalent amount of pAID. However, the other residues in the 112-bp hexasome were hardly changed upon the pAID addition (Supplementary Fig. 3b, d). Therefore, the signals of A126_i, K127_i, and G128_i are probably derived from the interaction of these residues with the free pAID displaced upon the addition of DNA. To clarify this point, we have added the following text and Supplementary Fig. 3d to the revised manuscript.

Line 371

“The irreconcilable low-field signals (A126_i, K127_i, and G128_i) in the 112-bp DNA/pAID nucleosome and 112/33-bp nucleosome at 0 mM NaCl roughly corresponded to those in the 112-bp hexasome after the addition of an equivalent amount of pAID (Supplementary Figs. 3d and 8d). However, the other residues in the 112-bp hexasome were hardly changed upon the addition of pAID (Supplementary Fig. 3b, d). This suggests that pAID partially interacts with Ala126, Lys127, and Gly128 of the H2A C-tail in the 112-bp DNA/pAID nucleosome. Probably, in the 112/33-bp nucleosome, the signals may derive from the interaction of these residues with free pAID that is displaced upon the addition of DNA.”

On the other hand, we apologize for the lack of sufficient data points in the DNA titration experiments with the 112-bp DNA/pAID nucleosome. We have now included HSQC spectra of the 112-bp DNA/pAID nucleosome upon titration with 33-bp DNA at molar ratios of 1:0.5 and 1:1 in Supplementary Fig. 2 of the revised manuscript. Significant signal changes were observed at Ser1, Arg3–Gly8, and Ala126–Gly128 of H2A and Lys16–Gly26 of H2B upon the addition of an equivalent amount of DNA (Supplementary Fig. 2c), and were roughly equivalent to those observed on two-fold addition of DNA (Supplementary Fig. 2d). This result suggests that the excess 33-bp DNA in the two-fold addition hardly affects the chemical shifts. To clarify this point,

we have added the following text and Supplementary Fig. 2 to the revised manuscript.

Line 137

“To determine which are the pAID side signals of the H2A and H2B tails, we titrated 33-bp DNA into the 112-bp DNA /pAID nucleosome incorporating labeled H2A and H2B at molar ratios of 0.5:1, 1:1, and 2:1 (Supplementary Fig. 2a–d). The signals of Ser1, Arg3–Gly8, and Ala126–Gly128 of H2A, and Lys16–Gly26 of H2B were significantly changed upon the addition of an equivalent amount of DNA (Supplementary Fig. 2c), and remained almost the same up to the two-fold addition of DNA (Supplementary Fig. 2d). This result suggests that the saturation point is more or less reached on equivalent addition, and the excess 33-bp DNA present after the two-fold addition of DNA hardly affects the chemical shifts.”

4. Question for Fig. 6, part b, lanes with FACT. On this type of gel, is it possible to distinguish between a hexasome-FACT complex and an assembled full nucleosome-FACT complex? It appears that all bands labeled as nucleosome-FACT complex run at the same place, but one of them is clearly a hexasome-FACT complex as no H2A-H2B dimer has been added in that lane (fourth from the right).

We appreciate the reviewer’s frank comment. More correctly, we note that the EMSA detected two bands as nucleosomal complexes with FACT in the lane of the 112-bp hexasome with FACT and an H2A/H2B dimer (Fig. 6b). To identify these nucleosomal complexes with FACT, we additionally performed a gel shift assay of the 112-bp hexasome upon the addition of an excess amount of FACT and the H2A/H2B dimer (Supplementary Fig. 10a). Based on the results, we concluded that these bands are not involved in the hexasome-FACT complex. To clarify this point, we have revised the following sentence, and added Supplementary Fig. 10a and the caption to the revised manuscript.

Line 446

“Note, however, that two bands corresponding to nucleosome-hFACT and nucleosome-hFACT-H2A/H2B were observed as slower bands (Fig. 6b and Supplementary Fig. 10a, b). In order to get the band intensity of the nucleosomal complexes with hFACT in Fig. 6c and Supplementary Fig. 10c, therefore, we added the band intensities of both the nucleosome-hFACT and nucleosome-hFACT-H2A/H2B complexes.”

Supplementary Fig. 10a

“Supplementary Figure 10. EMSAs of nucleosome assembly with FACT.

(a) Representative SYBR Gold-stained (left) and CBB-stained (right) EMSAs of nucleosomal complexes in different mixtures. The cartoon model of two nucleosomal complexes with FACT is shown as in Fig. 6a. Experiments were repeated at least three times. The full gel image is shown in Supplementary Fig. 13. For the 1:1 mixture of FACT and the 112-bp hexasome, the CBB-stained gel detected two main bands and a slower-migrating faint band corresponding to the 112-bp hexasome, FACT, and the complex with FACT, respectively. The hexasome band and the faint band of the complex were hardly changed upon the excess addition of FACT, indicating that FACT hardly binds to the 112-bp hexasome. On the other hand, for the 1:1:1

mixture of the 112-bp hexasome, FACT, and the H2A/H2B dimer, the band intensities of hexasome and FACT were significantly reduced, while two bands of complexes with FACT were more obvious. Upon the two-fold addition of H2A/H2B dimer to the 112-bp hexasome and FACT, the two bands corresponding to the nucleosomal complexes with FACT were clearly joined to the slower band, which is regarded as the nucleosome-FACT complex with the additional H2A/H2B dimer. Taken together, the two bands are the nucleosome-FACT and nucleosome-FACT-H2A/H2B complexes, respectively. Therefore, the band intensity of the nucleosomal complexes with FACT shown in Fig. 6c and Supplementary Fig. 10c were derived from the bands of both complexes.”

5. Also, for this assembly experiment, does the deletion of the H2A tail on its own impede assembly with FACT or the deletion of both H2A and H2B tails is required?

To investigate the effect of the H2A N-tail alone on nucleosome assembly with FACT, we additionally performed the nucleosome assembly experiment using a truncated mutant of the H2A N-tail (H2A Δ 9/H2B dimer) (Supplementary Fig. 10b). Deletion of the H2A N-tail hardly affected formation of the nucleosomal complexes with FACT, pAID, and DNA from the hexasome (Supplementary Fig. 10c). Therefore, deletion of the N-tails of both H2A and H2B (H2A Δ 9/H2B Δ 24 dimer) is required to impede the complex formation with pAID and FACT (Fig. 6b, c). We have corrected the text, and added Supplementary Fig. 10b, c to the revised manuscript, as follows.

Line 452

“In addition, deletion of the N-tail of either H2B (H2A/H2B Δ 24 dimer) (Fig. 6b, c) or H2A (H2A Δ 9/H2B dimer) (Supplementary Fig. 10b, c) hardly affected the formation of all three complexes; however, deletion of the N-tails of both H2A and H2B (H2A Δ 9/H2B Δ 24 dimer) significantly reduced complex formation with pAID and hFACT relative to complex formation with 33-bp DNA (Fig. 6b, c).”

Supplementary Fig. 10b, c

6. And finally, I found the paper hard to read and follow. It would have been helpful to summarize the observed changes in different spectra in a table format so that one can easily refer back to the relevant residues rather than search the text. Also, it might be helpful to add a section in the discussion or results, where each nucleosome type is compared with each other, e.g. 112 bp/pAID to wild-type, hexasome to wild-type, 33bp/112pb nucleosome to wild-type, hexasome to 112 bp/pAID etc., so that the reader can connect better with Fig. 7. I also had a really hard time distinguishing the colors in some figures, e.g. the red and yellow (?) in Figure 4.

We thank the reviewer for the helpful comment. In response, we have added Supplementary Tables 1 and 2 summarizing the changes of each signal at 100 mM NaCl in the revised manuscript, as indicated below. To improve the readability of our manuscript, we have also thoughtfully corrected the descriptions where histone tails in each nucleosomal complex are compared with each other. In addition, we have corrected the colors so that they can be distinguished easily in Fig. 3a, b and Supplementary Fig. 8a, b, d of the

revised manuscript.

Fig. 3a, b

Supplementary Fig. 8a, b, d

Supplementary Table 1. Summary of the observed signal changes of H2A in the NMR spectra of different complexes at 100 mM NaCl. Corresponding signals in each nucleosomal complex are displayed side by side.

H2A residues		112-bp DNA/pAID nucleosome	145-bp nucleosome	112-bp hexasome	Contact state
N-tail	Ser1	S1 _p	-	-	pAID
		S1 _d	S1	S1	DNA
	Gly2	G2	G2 _h	G2	Reduced DNA
		-	G2 _l	-	DNA
	Arg3	R3 _p	-	-	pAID
		R3 _{dh}	R3 _h	R3	DNA
		R3 _{dl}	R3 _l : Shift	-	DNA
	Gly4	G4	G4 _h	G4	Reduced DNA
		-	G4 _l	-	DNA
	Lys5	K5 _p	-	-	pAID
		-	K5 _h	-	DNA
		K5 _d	K5 _l	K5	Reduced DNA
	Gln6	Q6 _p	-	-	pAID
		Q6 _d	Q6 _h	Q6	Reduced DNA
		-	Q6 _l	-	DNA
	Gly7	G7 _p	-	-	pAID
G7 _d		G7 _h	G7	Reduced DNA	
-		G7 _l	-	DNA	
Gly8	G8 _p	-	-	pAID	
	G8 _d	G8	G8	DNA	
C-tail	Thr120	-	T120	-	-
	Glu121	E121	E121	E121	DNA/pAID
	Ser122	S122	S122	S122	DNA/pAID
	His123	-	-	-	-
	His124	-	-	-	-
	Lys125	K125	K125	K125	DNA/pAID
	Ala126	A126	A126	A126	DNA/pAID
	Lys127	K127 _h	K127	K127	DNA/pAID
		K127 _l	-	-	Free pAID
	Gly128	G128 _h	G128	G128	DNA/pAID
		G128 _l	-	-	Free pAID
	Lys129	K129 _h	K129	K129	DNA/pAID
K129 _l		-	-	Free pAID	

Supplementary Table 2. Summary of the observed signal changes of H2B in the NMR spectra of different complexes at 100 mM NaCl. Corresponding signals in each nucleosomal complex are displayed side by side.

H2B residues		112-bp DNA/pAID nucleosome	145-bp nucleosome	112-bp hexasome	Contact state
LS	Pro1	-	-	-	-
	Glu2	E2	E2	E2	DNA/pAID
	Pro3	-	-	-	-
	Ala4	A4	A4	A4	DNA/pAID
	Lys5	K5	K5	K5	DNA/pAID
	Ser6	S6	S6	S6 _h	DNA/pAID
		-	-	S6 _l	DNA/pAID
	Ala7	A7	A7	A7	DNA/pAID
	Pro8	-	-	-	-
	Ala9	A9	A9	A9	DNA/pAID
Pro10	-	-	-	-	
BS1	Lys11	K11	K11	K11	DNA/pAID
	Lys12	K12	K12	K12	DNA/pAID
	Gly13	G13	G13	G13	DNA/pAID
	Ser14	S14	S14	S14	DNA/pAID
	Lys15	K15	K15	K15	DNA/pAID
BS2	Lys16	K16 _p	-	-	pAID
		K16 _d	K16	K16	DNA
	Ala17	A17 _p	-	-	pAID
		A17 _d	A17 _h	A17 _h	DNA
		-	A17 _l	A17 _l	Reduced DNA
	Val18	V18 _p	-	-	pAID
		V18 _d	V18 _h	V18	DNA
		-	V18 _l	-	Reduced DNA
	Thr19	T19	T19 _h	T19	DNA
		-	T19 _l	-	Reduced DNA
	Lys20	K20	K20	K20	DNA
	Ala21	A21 _p	-	-	pAID
		A21 _d	A21 _h	A21	DNA
		-	A21 _l	-	Reduced DNA
	Gln22	Q22	Q22 _h	Q22	DNA
		-	Q22 _l	-	Reduced DNA
	Lys23	K23 _p	-	-	pAID
		K23 _{dh}	K23 _h	K23	DNA
		K23 _{dl}	K23 _l	-	DNA
	Lys24	K24	K24 _h	K24	DNA
-		K24 _l	-	Reduced DNA	
LS	Asp25	D25	D25	D25	DNA/pAID
	Gly26	G26	G26	G26	DNA/pAID
	Lys27	K27	K27	K27	DNA/pAID
Lys125	K125 _p	-	-	pAID	

Reviewers' comments:

Reviewer #1 (Remarks to the Author):

The authors have performed additional experiments and revised the manuscript that addressed all of my previous comments. I recommend publishing this work in Communications Biology.

Two minor issues:

1. Page 9, Line 297, Gly8d should be G8d.
2. Line 636, NMR spectrometry should be NMR spectroscopy.

Reviewer #2 (Remarks to the Author):

The authors have significantly improve the data and its discussion within the literature.

Reviewer #3 (Remarks to the Author):

The new version of the manuscript by Tsunaka et al. includes a large number of new supplementary figures, references, data, and more details regarding key results, conclusions and experiments. Overall, I found that the authors have addressed my major concerns and that the manuscript has been much improved.

I still have a few lingering questions about Figures 6 and 8:

Figure 6b:

- Lanes 3 and 7: These lanes contain 112bp hexasome + pAID or 33bp DNA, which should not give a full nucleosome band. However, both lanes have a band that looks consistent with a full nucleosome band. What is the origin of this band?
- Similarly, for lanes 5, 6, 9 and 10: Why is there a full nucleosome band here when truncated forms of H2A/H2B have been used?
- What is the origin of the topmost band in lanes 12, 13, and 14 (above the lanes labeled as FACT nucleosome/dimer complexes)?

Figure 8c:

- What is the evidence that the H2B N-tail binds towards the DNA entry-exit site?

Figure 8d:

- The expansion of the space between the DNA gyres presumably comes from the cryoEM model. This should be clarified and referenced.

All of our changes are marked in red in the revised manuscript. In this response letter, the reviewers' comments are indicated in black, and our responses are written in blue.

Reviewers' comments:

Reviewer #1 (Remarks to the Author):

The authors have performed additional experiments and revised the manuscript that addressed all of my previous comments. I recommend publishing this work in Communications Biology.

Two minor issues:

1. Page 9, Line 297, Gly8d should be G8d.
2. Line 636, NMR spectrometry should be NMR spectroscopy.

Thank you for your kind comment. We have corrected these words in the revised manuscript.

Reviewer #2 (Remarks to the Author):

The authors have significantly improve the data and its discussion within the literature.

Thank you for your appreciation of our revisions.

Reviewer #3 (Remarks to the Author):

The new version of the manuscript by Tsunaka et al. includes a large number of new supplementary figures, references, data, and more details regarding key results, conclusions and experiments. Overall, I found that the authors have addressed my major concerns and that the manuscript has been much improved.

I still have a few lingering questions about Figures 6 and 8:

We thank the reviewer for the comments. We have corrected in the revised manuscript, as described in detail in our point-by-point responses below.

Figure 6b:

1. Lanes 3 and 7: These lanes contain 112bp hexasome + pAID or 33bp DNA, which should not give a full nucleosome band. However, both lanes have a band that looks consistent with a full nucleosome band. What is the origin of this band?

Despite our intensive efforts in the reconstitution and purification of the 112-bp hexasome, we could not prevent the slight formation of the 112-bp nucleosome, comprising a histone octamer and 112-bp DNA. The 112-bp nucleosome seems to immediately interact with additional pAID or 33-bp DNA, thereby forming a faint band consistent with a full nucleosome band. Nevertheless, we consider that the presence of the faint band hardly affects our conclusion. To clarify this point, we have added the following text in the caption of Figure 6.

Caption of Figure 6b

“In lanes 3, 5, 6, 7, 9, and 10, a faint band consistent with a full nucleosome band is observed. The reconstitution of the 112-bp hexasome by the salt dialysis method could not prevent yielding a small quantity of the 112-bp nucleosome, comprising a histone octamer and 112-bp DNA. The faint band in lanes 3, 5, and 6 is likely to correspond to its complex with additional pAID, and the faint band in lanes 7, 9, and 10 may correspond to its complex with additional 33-bp DNA. Also, the faint band in lane 11 is likely to correspond to the 112-bp nucleosome interacting with hFACT.”

2. Similarly, for lanes 5, 6, 9 and 10: Why is there a full nucleosome band here when truncated forms of H2A/H2B have been used?

We consider that the faint band was detected for the same reason as mentioned in above response.

3. What is the origin of the topmost band in lanes 12, 13, and 14 (above the lanes labeled as FACT nucleosome/dimer complexes)?

It is possible that this faint band may originate from the slight formation of the 112-bp nucleosome/FACT/112-bp nucleosome complex upon the additions of the H2A/H2B dimer and FACT. However, it is unlikely that the presence of the band affect our conclusion.

Figure 8c:

4. What is the evidence that the H2B N-tail binds towards the DNA entry-exit site?

We have no direct evidence, but can suggest the model based on the NMR data. We have clarified this point in the revised manuscript, as follows.

Line 566:

“Thus, the H2B N-tail in hexasome seems to adopt one conformation bound to two DNA gyres in the direction toward the entry/exit site (Fig. 8d), which corresponds to a contact state to DNA in the BS2 region. In addition, the chemical shifts of the BS2 region in the 112-bp hexasome are roughly similar to those on the DNA side in the 112-bp DNA/pAID nucleosome (Fig. 5b). This suggests that the BS2 region on the DNA side in the 112-bp DNA/pAID nucleosome likely adopts the conformation toward the entry/exit site (Fig. 8c). This model is well consistent with the previous MD simulation that the H2A N-tail location on the major groove of DNA correlates with the H2B N-tail location toward the entry/exit site in the canonical nucleosome³³ (Fig. 8b).”

Figure 8d:

5. The expansion of the space between the DNA gyres presumably comes from the cryoEM model. This should be clarified and referenced.

Thank you for the kind comment. To clarify this point, we have added the following text in the caption of Figure 8.

Caption of Figure 8

“Black arrow represents expansion of the space between two DNA gyres in hexasome, as shown in the cryo-EM structure of the 112-bp hexasome¹⁰.”